# MME+ fibro-adipogenic progenitors are the dominant adipogenic population during fatty infiltration in human skeletal muscle

Gillian Fitzgerald[1,8], Guillermo Turiel[1,8], Tatiane Gorski[1], Inés Soro-Arnaiz[1], Jing Zhang[1], Nicola C. Casartelli[1,2], Evi Masschelein[1], Nicola A. Maffiuletti[2], Reto Sutter[3,4], Michael Leunig[5], Jean Farup[6,7] & Katrien De Bock [1✉]

Fatty infiltration, the ectopic deposition of adipose tissue within skeletal muscle, is mediated via the adipogenic differentiation of fibro-adipogenic progenitors (FAPs). We used single-nuclei and single-cell RNA sequencing to characterize FAP heterogeneity in patients with fatty infiltration. We identified an MME+ FAP subpopulation which, based on ex vivo characterization as well as transplantation experiments, exhibits high adipogenic potential. MME+ FAPs are characterized by low activity of WNT, known to control adipogenic commitment, and are refractory to the inhibitory role of WNT activators. Using preclinical models for muscle damage versus fatty infiltration, we show that many MME+ FAPs undergo apoptosis during muscle regeneration and differentiate into adipocytes under pathological conditions, leading to a reduction in their abundance. Finally, we utilized the varying fat infiltration levels in human hip muscles and found less MME+ FAPs in fatty infiltrated human muscle. Altogether, we have identified the dominant adipogenic FAP subpopulation in skeletal muscle.

[1] Laboratory of Exercise and Health, ETH Zurich, Schwerzenbach, Switzerland. [2] Human Performance Lab, Schulthess Clinic, Zurich, Switzerland. [3] Department of Radiology, University Hospital Balgrist, Zurich, Switzerland. [4] Faculty of Medicine, University of Zurich, Zurich, Switzerland. [5] Department of Orthopaedic Surgery, Schulthess Clinic, Zurich, Switzerland. [6] Department of Biomedicine, Aarhus University, Aarhus, Denmark. [7] Steno Diabetes Center Aarhus, Aarhus University Hospital, Aarhus, Denmark. [8] These authors contributed equally: Gillian Fitzgerald, Guillermo Turiel. ✉email: katrien-debock@ethz.ch

The ectopic deposition of intramuscular adipose tissue, termed fatty infiltration or fatty degeneration, is a hallmark of aging and contributes to the pathological progression of various myopathies. Fatty infiltration has been observed in a wide range of diseases and clinical conditions, including insulin resistance and diabetes[1–3], primary muscular dystrophies[4,5], muscle wasting due to aging or disease[5–7], as well as incomplete muscle repair after injury and/or surgical interventions[8,9]. Fatty infiltration is associated with decreased physical capacity and impaired metabolic function in skeletal muscle pathologies[2,10,11], and correlates with disease severity and poor outcome under many conditions[12–14], but there are currently no pharmacological strategies to reverse or prevent it.

About a decade ago, skeletal muscle resident fibro-adipogenic progenitors (FAPs) were identified and characterized as stem cell antigen (SCA) 1+CD34+platelet-derived growth factor receptor α (PDGFRα)+ cells. These cells exhibit fibrogenic and adipogenic properties and contribute to muscle fatty infiltration after glycerol injection in murine muscle in vivo[15,16]. However, their role is contextual. In healthy muscle, PDGFRα+ FAPs are required for the maintenance of skeletal muscle homeostasis. They also contribute to muscle repair by generating a transitional niche which supports the activation and differentiation of muscle stem cells[15,17–20]. The number and function of FAPs is tightly controlled by interactions with other cell types in the muscle niche. For example, immune cells have been shown to stimulate FAP proliferation shortly after muscle damage but at a later stage directly induce FAP apoptosis[21,22]. Although we are now starting to understand the physiological function of FAPs in the support of myogenesis during normal regeneration, the specific molecular mechanisms underlying their differentiation into adipocytes under pathological settings, including chronic muscle degeneration, remain poorly understood. In particular, while adipogenic differentiation in mouse models is often induced by glycerol injection and/or is correlated with impaired regeneration, when and how FAPs undergo adipogenic differentiation in humans is not known.

Recent single-cell RNA sequencing (scRNA-seq) approaches have underlined the complexity and heterogeneity of FAPs and identified the existence of several FAP subpopulations in healthy human skeletal muscle[23,24]. The specific role and contribution of FAP subpopulations to muscle repair and/or fibro-fatty degeneration was only recently suggested when studies revealed an increase in the presence of THY1+ (CD90+) FAPs in type 2 diabetic patients[1]. CD90+ FAPs exhibit high proliferative capacity, clonogenicity, and extracellular matrix (ECM) production, which is associated with degenerative remodeling of the ECM and fibrosis in diabetic patients. Thus, studying FAP heterogeneity in humans might lead to novel insights into the contribution of FAPs to muscle pathologies.

In this study, we used single-nuclei RNA sequencing (snRNA-seq) and scRNA-seq to characterize FAP subpopulations in skeletal muscle from patients undergoing total hip arthroplasty (THA) and identified an MME+ FAP subpopulation with a unique adipogenic signature. We characterized this population using ex vivo techniques in combination with transplantation experiments to show that it exhibits a high potential for adipogenic differentiation. MME+ FAPs are characterized by low activity of WNT signaling, a pathway known to repress adipogenesis, and are refractory to the inhibitory role of WNT activators. Furthermore, using cardiotoxin and glycerol injections as preclinical models for muscle regeneration and fatty degeneration respectively, we showed that many MME+ FAPs undergo apoptosis during muscle regeneration and differentiate into adipocytes upon glycerol injection, leading to their depletion. Finally, by taking advantage of different fat infiltration levels in human hip muscles, we confirmed the depletion of MME+ FAPs in fatty infiltrated human muscle. Altogether, we have identified MME+ FAPs as the dominant adipogenic FAP subpopulation in skeletal muscle.

## Results

**snRNA-seq in human skeletal muscle identifies 3 FAP subtypes.** To explore the cellular origins of human muscle fatty infiltration, we harvested muscles from patients ($n = 5$) undergoing total hip arthroplasty (THA) due to hip osteoarthritis (HOA), a condition associated with muscle fatty infiltration[25,26]. To include adipocytes, which float and are generally lost in single-cell preparations due to their high lipid content, we opted for single-nuclei RNA sequencing (snRNA-seq)[27]. After tissue dissociation and nuclei extraction, we used fluorescence activated cell sorting (FACS) to enrich the sample with Hoechst+ nuclei and used 10X Chromium technology to perform snRNA-seq. For the bioinformatic analysis, we selected high quality nuclei based on three parameters: library size, number of expressed genes, and proportion of reads mapped to mitochondrial genes. For the latter parameter, we detected an average of 0.05% mitochondrial genes per nuclei indicating a high purity of the sample. After quality control filtering, we obtained a dataset of 26981 nuclei with an average of 2085 genes per nuclei. Expression values were log-normalized and we performed unsupervised clustering and marker gene detection to identify the main cell populations in the dataset (Supplementary Fig. 1a, b). The dataset was mainly composed of myonuclei ($\approx$80%), similar to other observations[28], but also of common muscle mononuclear cell populations such as FAPs, immune cells, endothelial cells, satellite cells and adipocytes. As the original patient samples ($n = 5$) were pooled together, we applied a demultiplexing method[29] that allows the assignment of the donor of origin for every cell based on single nucleotide polymorphism (SNPs) differences in the sequencing data. As shown in Supplementary Fig. 1c, d, the five different donors contributed to all clusters indicating that these populations are common for all patients, and therefore, they are not a consequence of patient variability. For the purpose of this manuscript, we narrowed our dataset to FAPs and adipocytes, since those cells are likely involved in fatty infiltration in human skeletal muscle[16]. We obtained 2591 nuclei containing FAPs and adipocytes with an average of 1899 genes per nuclei. They expressed well-known marker genes: FAPs were marked by expression of *PDGFRA*, *CD34*, and *DCN* whereas adipocytes displayed expression of marker genes such as *PLIN1*, *PPARG*, and *ADIPOQ*. Clustering analysis identified three different FAP populations and one adipocyte population (Fig. 1a, b). These FAP populations were characterized by the differential expression of several marker genes which confirms that the clustering finely resolved the main populations of the dataset (Fig. 1c and Supplementary Fig. 2a). Also in this dataset, SNPs-based demultiplexing showed that all patients contributed to the 3 FAPs and adipocytes populations indicating that they are not a consequence of patient variability (Supplementary Fig. 2b, c). To gain further insight into the differences between the FAP subpopulations, we performed a Gene Set Enrichment Analysis (GSEA) using the Hallmark pathways from MsigDB[30]. This collection of curated gene sets comprises 50 different pathways which summarize well-defined biological states such as hypoxia, angiogenesis, adipogenesis, and metabolic processes, among others. We performed pairwise comparisons between all the FAP subpopulations and found a consistent enrichment of adipogenesis in FAPs 1 compared to either FAPs 2 or FAPs 3 populations (Fig. 1d and Supplementary Fig. 2d). We also used Gene Set Variation Analysis (GSVA)[31] which calculates a score per cell based on the

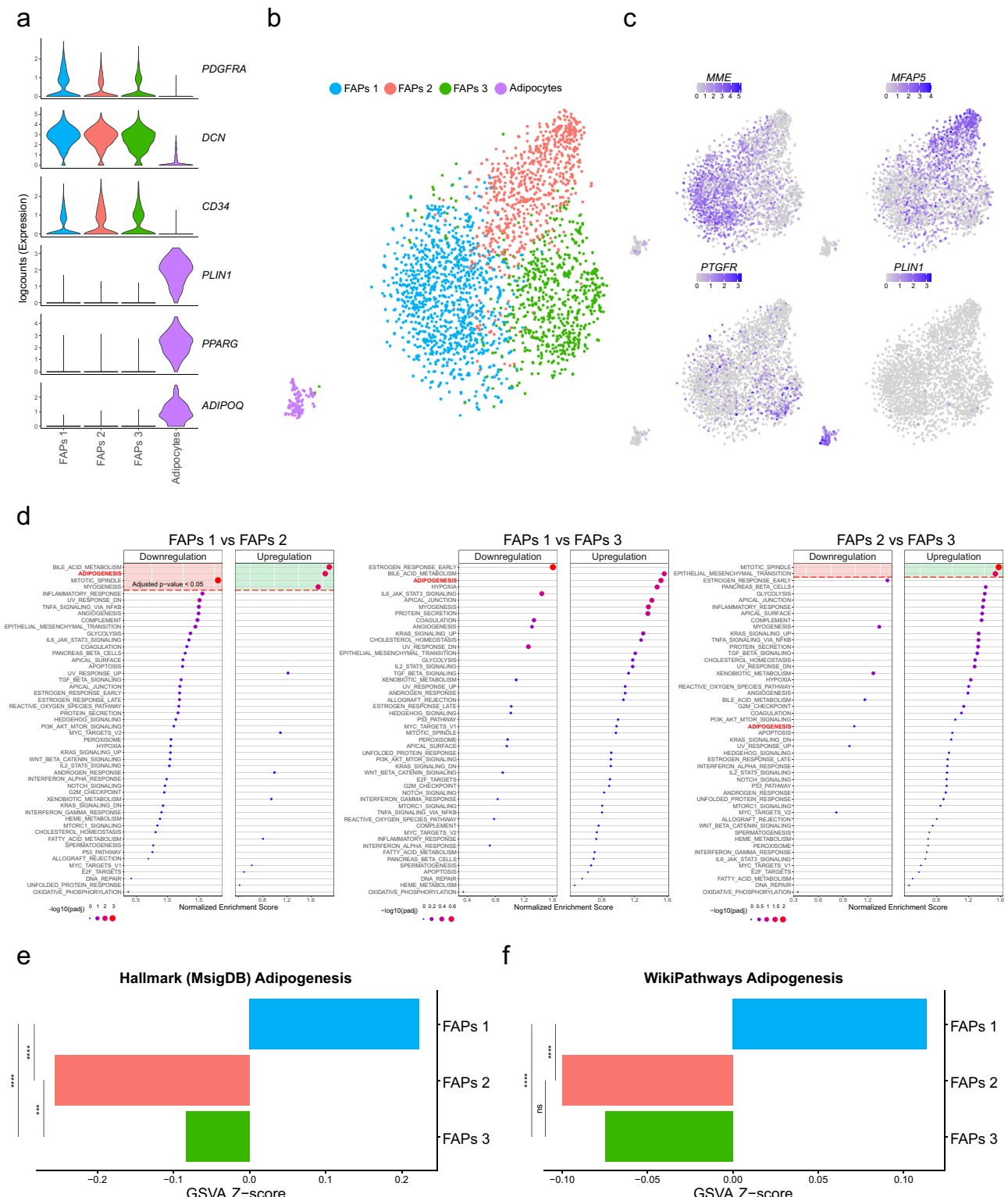

**Fig. 1 snRNA-seq analysis from human muscle identifies 3 FAP populations. a** Violin plots of logcounts values from well-known marker genes of FAPs (*PDGFRA*, *DCN* and *CD34*) and adipocytes (*PLIN1*, *PPARG* and *ADIPOQ*), color-coded by the identified subpopulations. **b** TSNE plot of FAP and adipocyte populations from human muscle, color-coded by the identified subpopulations. **c** TSNE plots of expression levels of marker genes from the different subpopulations, color-coded by logcounts values. **d** Dot plots showing upregulated or downregulated pathways from GSEA of FAPs 1 vs FAPs 2 (left panel), FAPs 1 vs FAPs 3 (middle panel) or FAPs 2 vs FAPs 3 (right panel). X-axis indicates the normalized enrichment score (NES) for each pathway. Color and size of the dots indicate adjusted *p*-values (FDR). Significant values are delimited by the red dashed line (Adjusted *p* < 0.05 or -log₁₀(adjusted *p*-value) >1.3). **e**, **f** Bar plots showing average Z-score of GSVA of adipogenesis pathways from Hallmark (MsigDB) (**e**) and WikiPathways databases (**f**). Color-coded by identified FAPs subpopulations. ***$p < = 0.001$, ****$p < = 0.0001$.

variation of pathway activity over a population. We applied this method with the adipogenesis pathways from Hallmark and WikiPathways databases[32]. In both cases, FAPs 1 showed a significant positive GSVA z-score when compared to the other two FAPs clusters (Fig. 1e, f) indicating a higher activity of these adipogenic pathways in FAPs 1. Based on these results, we raised the question whether the FAP subpopulations in human skeletal muscle could harbor different adipogenic potential.

To validate the results from the snRNA-seq analysis, as well as to overcome the inherent issue of recovering a high proportion of myonuclei from muscle tissue, we performed single-cell RNA sequencing (scRNA-seq) in human skeletal muscle. We harvested a sample from the *rectus femoris* muscle of a patient undergoing THA. We isolated viable (7AAD⁻) and metabolically active (calcein⁺) mononuclear cells and used 10X Chromium technology to perform scRNA-seq. After quality control filtering, we obtained 6256 cells with an average of 1553 genes per cell. Expression values were log-normalized, and we then applied an unsupervised clustering and marker gene detection to identify the main cell populations that constitute the dataset (Fig. 2a, b and Supplementary Fig. 3a, b). Based on the differential expression of known marker genes, we could detect several immune cell populations (NK/T/B cells and macrophages), mural cells (smooth muscle cells/pericytes), endothelial cells, satellite cells and 3 FAP populations. We subsequently labeled these 3 FAP populations by top surface marker genes: *MME*⁺, *CD55*⁺, and *GPC3*⁺ FAPs (see below) (Fig. 2a, b and Supplementary Fig. 3a, b). As expected, adipocytes were not detected in this experiment. We subsequently compared our dataset to two recently published scRNA-seq datasets in human skeletal muscle[23,24]. To do so, we used an unbiased method to automatically annotate our data using reference datasets[33]. This method compares both samples and assigns the labels (e.g., cell types) from the reference dataset to the cells of our data based on Spearman rank correlations of expression profiles. Mapping to both Rubenstein et al. and De Micheli et al. datasets resulted in clearly defined populations (Supplementary Fig. 3c, d) and a high correlation between their annotation and our original clustering (Fig. 2c, d) therefore confirming the accuracy of this classification. One classification which did not fully agree with our original clustering was the mapping of our 3 FAP populations to the population labelled as "adipocytes" by De Micheli et al. (Fig. 2d and Supplementary Fig. 3d). However, we continue to classify these three populations in our dataset as FAPs due to their expression of FAP marker genes such as *PDGFRA*, *CD34*, and *DCN* and lack of expression of mature adipocyte marker genes such as *PPARG* and *ADIPOQ* (Fig. 2b).

We subsequently more closely investigated the FAP populations in our human dataset. These 3 FAP populations accounted for 3622 of our total of 6256 cells (57.9%). Those populations were characterized by the ubiquitous expression of the FAP markers *PDGFRA*, *CD34*, and *DCN* along with the lack of *NCAM1*(*CD56*) expression[16,34] (Fig. 2b), but could be differentiated as distinct cell populations through their expression of unique sets of differentially expressed genes (Fig. 2e and Supplementary Fig. 3a, b). The *CD55*⁺ FAP population was defined by a marked differential expression of *TNXB*, *MFAP5*, *PCOLCE2*, *FBN1*, *CD55* and *PRG4*; each of which have been associated with synovial cell and/or chondrocyte expression[35–40]. Their gene expression pattern is very similar to the *PRG4*⁺ FAP population described by Rubenstein et al. as well as the *FBN1*⁺*MFAP5*⁺*CD55*⁺ fibroblasts population described by De Micheli et al. (Fig. 2c–e and Supplementary Fig. 3a, c, d). Second, we found an *MME*⁺ FAP population which was marked by expression of a different set of genes, many of which are involved in vascular processes. *MME* (*CD10*), one of the most differentially expressed gene in this cluster, encodes a matrix

metalloendopeptidase that cleaves a variety of peptide hormones including Substance P, involved in vasodilation[41,42]. Furthermore, *PTGDS*, *CXCL14*, and *SMOC2* were also among the top differentially expressed genes in the *MME*⁺ FAP population, each of which has been shown to play a role in angiogenesis[43–45]. This population closely resembled *LUM*⁺ FAPs as well as *MYOC*⁺ and *APOD*⁺*CFD*⁺*PLAC9*⁺ populations from Rubenstein et al. and De Micheli et al., respectively (Fig. 2c–e and Supplementary Fig. 3a, c, d). Lastly, *GPC3*⁺ FAPs appeared to be an intermediate population with marker genes overlapping with both *MME*⁺ and *CD55*⁺ FAP populations along with differentially expressed genes including *GPC3* and *SFRP2*, both involved in WNT signaling[46,47]. In both comparisons, to Rubenstein et al. and De Micheli et al., this population aligned within the same population as MME⁺ cells (Fig. 2c–e and Supplementary Fig. 3a, c, d). While this may indicate a closer relationship between *GPC3*⁺ FAPs and *MME*⁺ FAPs, its overlapping expression patterns with both *MME*⁺ as well as *CD55*⁺ FAP populations, along with unique expression of various other genes, warranted their definition as an independent 3rd FAP population (Fig. 2e and Supplementary Fig. 3a).

Finally, we compared the 3 FAPs populations identified in the scRNA-seq with those from the snRNA-seq to confirm that we detected the same FAP subtypes in both RNA-seq experiments. We selected the top marker genes for each FAP population from the nuclei data and performed a Pearson's correlation analysis in the scRNA-seq data. Interestingly, the set of marker genes from FAPs 1, FAPs 2 and FAPs 3 had a high and specific correlation with *MME*⁺, *CD55*⁺ and *GPC3*⁺ FAPs, respectively (Supplementary Fig. 3e). Furthermore, we performed the same Pearson's correlation analysis with the top marker genes from the scRNA-seq data over the nuclei data and we detected the same high and specific correlation profile (Supplementary Fig. 3f) indicating that we detected the same three FAP populations with each technology.

**MME and GPC3 mark FAP subtypes not found in human adipose stromal populations**. Similar to adipose-derived progenitor cells, FAPs widely express mesenchymal progenitor markers such as PDGFRα and CD34 and display a high adipogenic potential. The contribution of progenitor cell populations to adipocytes has been intensely studied in adipose tissue and has led to the discovery of a *DPP4*⁺ interstitial progenitor cell population that gives rise to committed *ICAM1*⁺ preadipocytes[48,49]. Given the broad similarities between muscle FAPs and adipose tissue progenitors, we decided to compare our dataset with a published dataset which profiled CD45-depleted stromal vascular cells from human abdominal subcutaneous white adipose tissue (scWAT)[48]. To do so, we batch-corrected and integrated the adipogenic progenitors from their single cell adipose tissue sample with our muscle sample using Harmony[50]. Interestingly, we found that *CD55*⁺ FAPs completely integrated with the *DPP4*⁺ interstitial progenitor cells while the *GPC3*⁺ and *MME*⁺ FAPs were clearly separated in independent populations and did not overlap with *ICAM1*⁺ cells (Fig. 3a). To gain further insight into this, we more closely investigated the transcriptomic profiles of the different cell populations. For each individual sample, we identified the marker genes that define the populations and then compared them to analyze how related they are. Consistent with the integration results, *DPP4*⁺ and *CD55*⁺ cells shared a large proportion of their marker genes (33/46 of *CD55*⁺ FAP marker genes), including, *DPP4*, *CD55*, *WNT2*, and *PI16* (Fig. 3b and Supplementary Fig. 4a). On the other hand, *ICAM1*⁺ cells only had limited overlap with *MME*⁺ FAPs (6/27 of *MME*⁺ FAP marker genes) as well as *GPC3*⁺ FAPs (6/29 of *GPC3*⁺ FAP marker genes) (Fig. 3b). Consistently, the marker genes related to

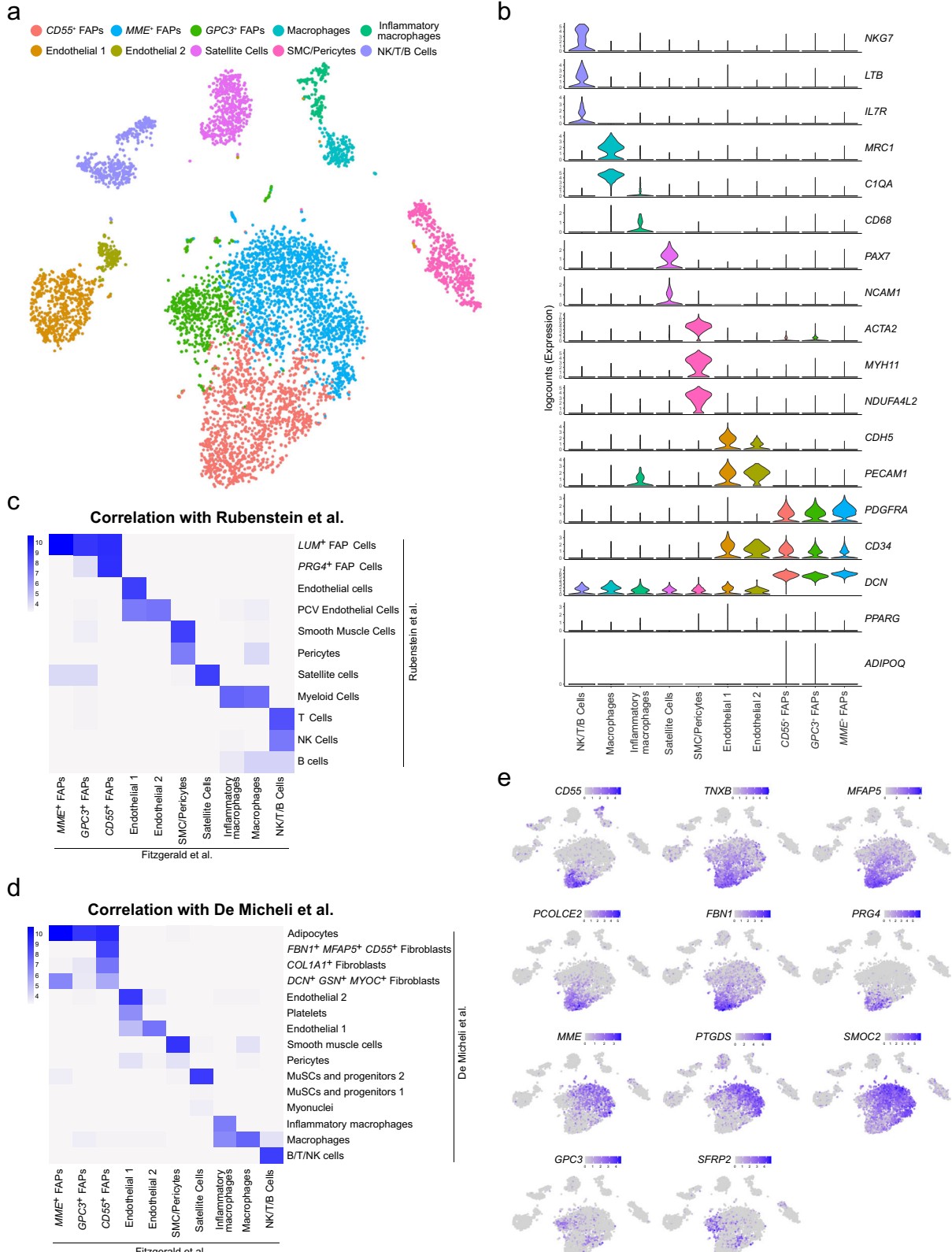

**Fig. 2 scRNA-seq analysis from human muscle identifies main FAP and mononuclear cell populations. a** TSNE plot of main cell types from human muscle, color-coded by the identified subpopulations. **b** Violin plots of logcounts values from well-known marker genes in each cell type, color-coded by the identified subpopulations. **c**, **d** Heatmap of the distribution of cells across Rubenstein et al. (**c**) and De Micheli et al. (**d**) labels (Y-axis) and the original clustering of the muscle scRNA-seq dataset (X-axis). Color scales represent the log10-number of cells for each label-cluster combination. **e** TSNE plots of expression levels of marker genes from the different FAPs subpopulations, color-coded by logcounts values.

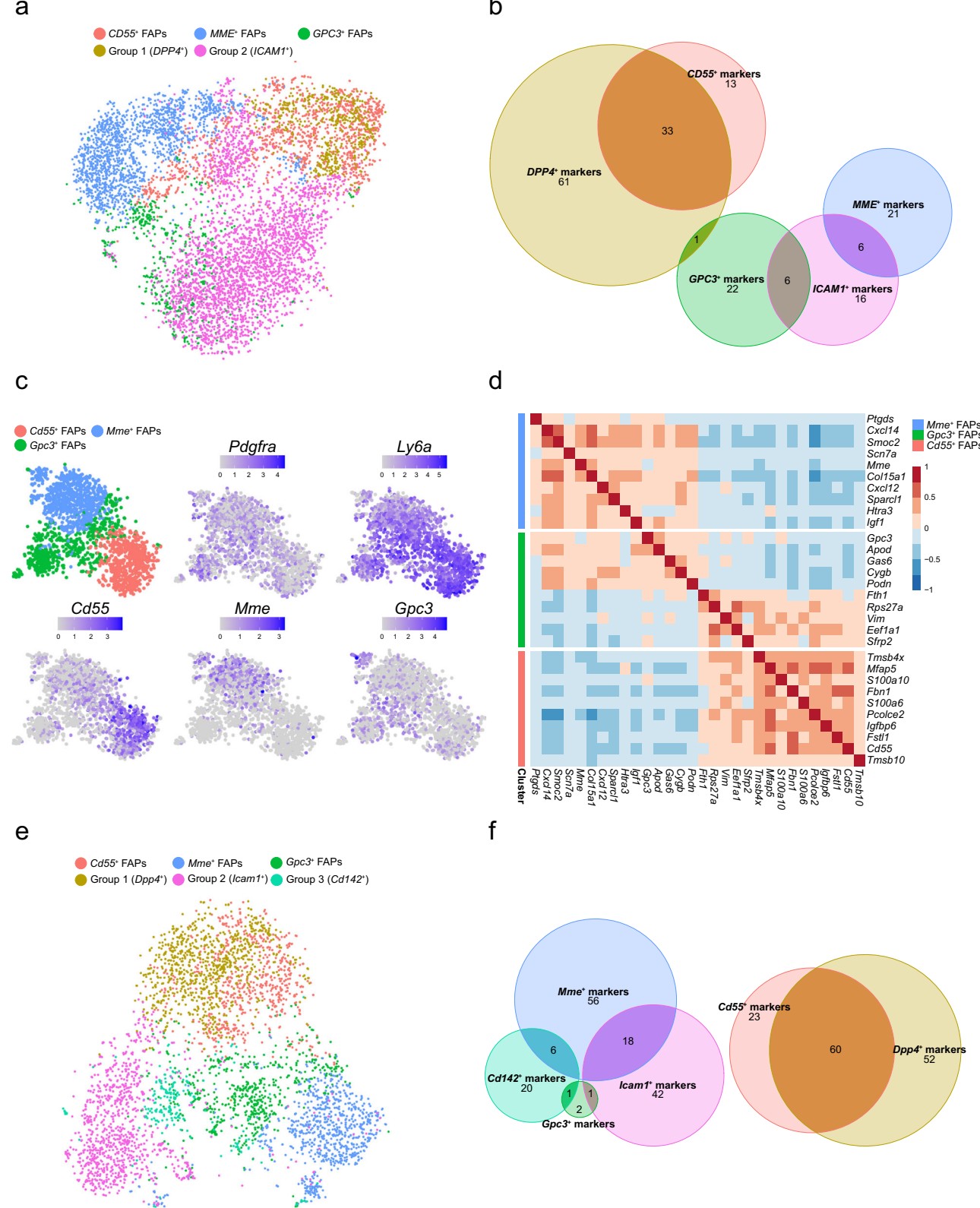

*ICAM1*[+] cells such as *MRPS6* and *AADAC* (*ICAM1* and *VCAM1* showed lower expression levels in scRNAseq), were not expressed in any of our muscle FAP populations (Supplementary Fig. 4a). Moreover, some of the most highly differentially expressed genes marking our *MME*[+] FAPs were not expressed in either *DPP4*[+] or *ICAM1*[+] cells from subcutaneous WAT including *MME*, *PTGDS* and *SCN7A* (Supplementary Fig. 4a). In conclusion, we found

that *CD55*[+] FAPs and *DPP4*[+] interstitial progenitors are closely related, while *MME*[+] and *GPC3*[+] FAPs show significant transcriptomic differences when compared to *ICAM1*[+] cells that separate them into distinct populations. We aimed to get further insight into the consequences of these differences at a pathway level. For this reason, we performed a GSEA using the Hallmark pathways from MsigDB comparing *MME*[+] or *ICAM1*[+] cells with

**Fig. 3 *MME*$^+$ and *GPC3*$^+$ muscle FAPs are not found in adipose tissue. a** TSNE plot of the integration of human muscle FAPs and human adipose progenitor cells from Merrick et al., color-coded by the original annotation of each dataset. **b** Euler diagram showing the number of common marker genes between human muscle FAPs and human adipose progenitor cells from Merrick et al. Marker genes were identified for each dataset independently. **c** TSNE plots of main FAP populations identified in mouse muscle (upper-left panel, color-coded by the identified populations), well-known marker genes for mouse FAPs (upper-middle and upper-right panels, color-coded by logcounts values) and marker genes for each FAP subpopulations (bottom panels, color-coded by logcounts values). **d** Heatmap representing Pearson's correlation values of human FAP markers in the mouse muscle scRNA-seq dataset. Horizontal color-coded separation indicates the set of marker genes from each FAP subpopulation. **e** TSNE plot of the integration of mouse muscle FAPs and adult mouse (10 weeks) adipose progenitor cells from Merrick et al., color-coded by the original annotation of each dataset. **f** Euler diagram showing the number of common marker genes between mouse muscle FAPs and adult mouse (10 weeks) adipose progenitor cells from Merrick et al. Marker genes were identified for each dataset independently.

---

*CD55*$^+$ or *DPP4*$^+$ cells respectively. Interestingly, both *MME*$^+$ and *ICAM1*$^+$ cells showed an upregulation in adipogenesis-related genes when compared to their associated *CD55*$^+$ FAPs or *DPP4*$^+$ interstitial progenitor subpopulation (Supplementary Fig. 4b). Surprisingly, we found a considerably different set of adipogenic genes upregulated either in *MME*$^+$ or *ICAM1*$^+$ cells (Supplementary Fig. 4b). These results further confirm the pronounced transcriptomic differences between *MME*$^+$ and *ICAM1*$^+$ cells but nonetheless suggest that these cells are the most highly adipogenic fraction of cells in their respective tissues.

Finally, to ensure that the differences observed between *MME*$^+$ and *ICAM1*$^+$ populations are not due to batch effects of samples coming from different laboratories, we integrated and compared FAPs coming from abdomen muscle versus subcutaneous adipose tissue generated within Tabula Sapiens[51]. This confirmed that both muscle and adipose tissues share a common *DPP4*$^+$*CD55*$^+$ population (Supplementary Fig. 4c), while the muscle-derived *MME* population (marker genes *MME*, *PTGDS* and *SCN7A*) clustered separately from the adipose tissue 'ICAM1$^+$' FAPs, which were characterized by *MRPS6* and *AADAC*. Therefore, the *MME*$^+$ FAP population we identified in human muscle is transcriptionally different from other FAPs populations present in human adipose tissue.

**Mouse FAP subtypes display similar transcriptomic properties as human FAP subtypes.** To further study muscle FAP populations, we performed an additional scRNA-seq experiment on mouse muscle to confirm that the different FAP populations detected in human samples are also present in mouse muscle. Following the workflow described above, we selected 2295 high quality cells with an average of 1923 genes identified per cell. Expression values were normalized, log-transformed and highly variable genes were selected based on quantification of per-gene variation. PCA, dimensionality reduction via t-SNE and clustering was performed the same way as the human data analysis. After clustering, 3 distinct FAP populations were identified based on the expression of the canonical FAP markers *Pdgfra* and *Ly6a* (*Sca-1*) (Fig. 3c). Interestingly, these mouse FAP populations were characterized by the expression of *Cd55*, *Mme* and *Gpc3* as in the human data (Fig. 3c and Supplementary Fig. 4d). To validate these populations with the human FAP dataset, we performed a Pearson's correlation analysis for the top marker genes of the human FAP populations over the mouse dataset (Fig. 3d). Human marker genes for each population were consistent in the mouse FAP subtypes. In agreement with our human dataset, marker genes for the *Gpc3*$^+$ FAP cells in mouse had a mixed correlation with the other two FAP populations further demonstrating that this population represents an intermediate state between *Mme*$^+$ FAP cells and *Cd55*$^+$ FAP cells. Similar to the human data, *Cd55*$^+$ FAPs expressed synovial/chondrocyte related genes such as *Mfap5*, *Pcolce2*, *Fbn1* and *Cd55*. *Gpc3*$^+$ FAPs showed expression of Wnt related genes including *Gpc3*. *Mme*$^+$ FAPs showed expression of genes related to vascular processes such as *Mme*, *Cxcl14* and *Smoc2* (Supplementary Fig. 4d). Based

on these results, we confirmed that the FAPs populations identified in human muscle are also present in mouse muscle.

We also integrated our mouse data with those from adipose progenitor cells isolated from adult mouse scWAT[48]. Again, we found that *Cd55*$^+$ FAPs integrated very well with *Dpp4*$^+$ progenitors identified by Merrick et al.[48] while the *Cd142*$^+$ and *Icam1*$^+$ committed pre-adipocytes substantially separated from the *Gpc3*$^+$ and *Mme*$^+$ FAPs (Fig. 3e). *Cd55*$^+$ FAPs from mouse expressed many of the *Dpp4*$^+$ progenitor marker genes (60/83 of *Cd55*$^+$ FAP marker genes were *Dpp4*$^+$ marker genes) such as *Dpp4*, *Cd55* and *Pi16* (Fig. 3f and Supplementary Fig. 4e). Consistent with this, CD55 was recently used as a marker to sort P1 progenitors in adipose tissue[52]. The marker genes for *Icam1*$^+$ cells such as *Icam1* as well as *Cd36*, *Fabp4* and *Pparg*, all of which are considered pre-adipogenic markers, were not expressed in *Gpc3*$^+$ and *Mme*$^+$ FAPs, (Supplementary Fig. 4e). Similarly, marker genes specific for the *Mme*$^+$ cells such as *Crispld2*, *G0s2*, *Atp1a2* and *Vwa1* were not expressed in *Cd142*$^+$ and *Icam1*$^+$ pre-adipocyte populations (Supplementary Fig. 4e). These expression profiles further confirm our observations in human samples where *Cd55*$^+$ FAPs and *Dpp4*$^+$ adipogenic progenitor cells are transcriptionally similar, while *Gpc3*$^+$ and *Mme*$^+$ are different from the (*Cd142*$^+$ and) *Icam1*$^+$ pre-adipocytes.

To gain some insight into the fate of *Mme*$^+$ FAPs in vivo, we also used a published scRNAseq dataset of muscle FAPs 5 days after glycerol injection[53] (5 dpi FAPs), integrated it with our mouse muscle FAPs dataset in basal condition and performed pseudotime analysis. First, at the end of the trajectory, we found a clear upregulation of adipocytes markers (*Plin1*, *Adipoq*, *Pparg*) in 5 dpi FAPs, underscoring their ongoing differentiation at 5 dpi (Supplementary Fig. 4f). Second, we observed that, when compared to *Cd55*$^+$ and *Gpc3*$^+$ FAPs, *Mme*$^+$ FAPs positioned closer to 5 dpi FAPs in the trajectory (Supplementary Fig. 4f). Interestingly, we found an increase in *Cebpa* expression, a transcription factor well known to induce adipogenesis along with *Pparg*[54], along the trajectory with the highest peak corresponding to *Mme*$^+$ FAPs (Supplementary Fig. 4f, g). Given the differential and unique adipogenic signature of the *Mme*$^+$ FAP population, we thus decided to study whether *Mme*$^+$ FAPs are the main population which contribute to fatty infiltration in skeletal muscle.

**MME$^+$ FAPs are a highly adipogenic fraction of PDGFRα$^+$ FAPs.** To study the functional properties of different muscle FAP populations, we subsequently isolated MME$^+$ and MME$^-$ FAPs from resting adult mouse muscle. MME is a cell surface enzyme which allowed us to isolate MME$^+$ cells via FACS. We used PDGFRα-eGFP mice, which express H2B-eGFP under the endogenous PDGFRα promoter to confirm the presence of PDGFRα$^+$(eGFP$^+$) MME$^+$ cells (MME$^+$ FAPs) and eGFP$^+$ MME$^-$ cells (MME$^-$ FAPs) via flow cytometry as well as immunofluorescent staining (Supplementary Fig. 5a, b). qPCR analysis on marker genes from MME$^+$ FAPs (*Mme* and *Lum*) as well as

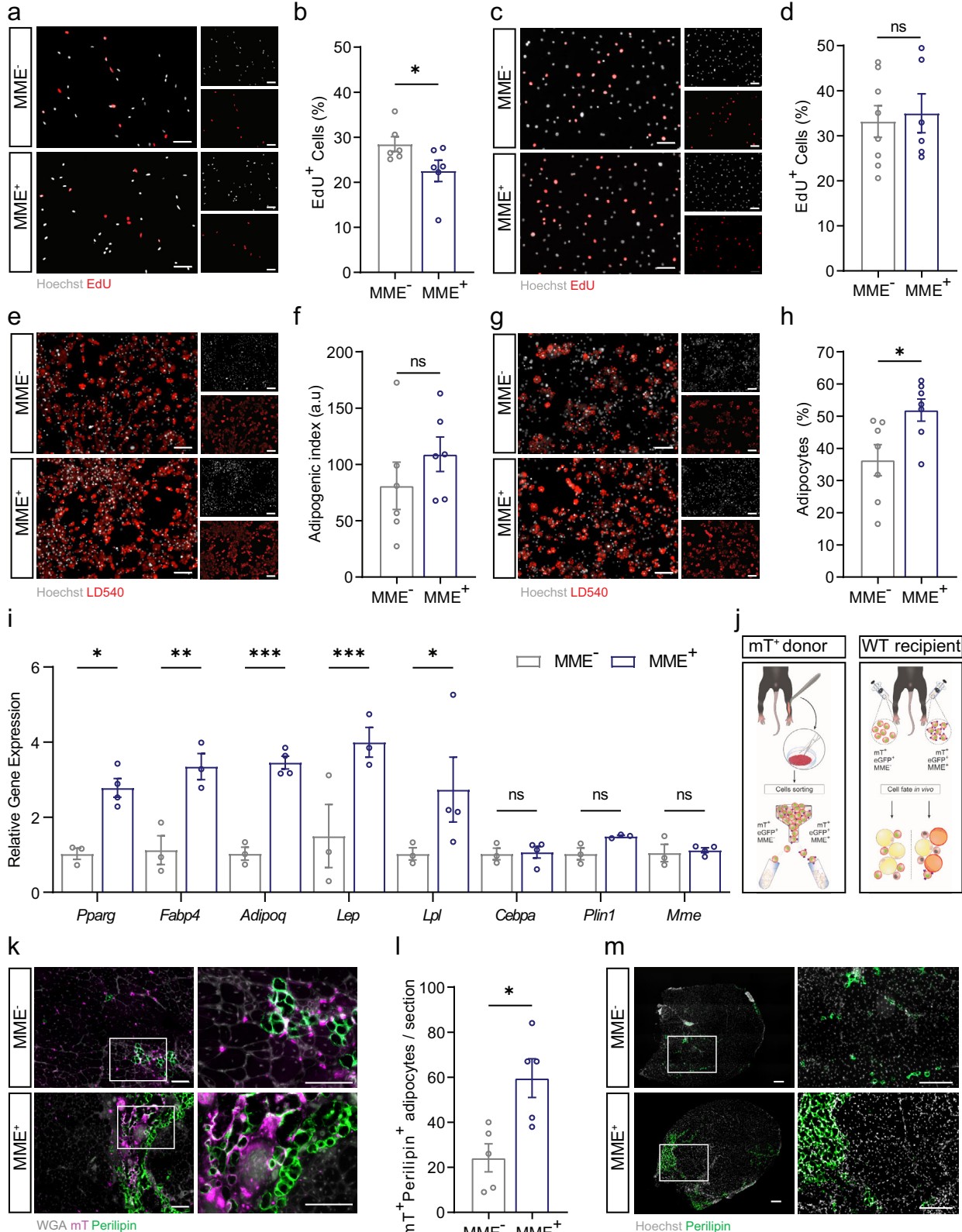

from CD55[+] FAPs (*Cd55* and *Fbn1*) confirmed a strong upregulation of the MME[+] FAP marker genes in MME[+] FAPs (Supplementary Fig. 5c, d) while CD55[+] FAP marker genes were more prominently expressed in MME[-] FAPs (Supplementary Fig. 5e, f). We initially noticed a small delay in cell cycle entry of MME[+] FAPs, as indicated by fewer cells incorporating EdU in the first 48 h in culture (Fig. 4a, b), but proliferation rates under standard culture conditions were not different between the two cell populations (Fig. 4c, d). Next, we investigated their differentiation potential into adipocytes. Both MME[+] and MME[-] FAPs underwent robust adipocyte differentiation under stimulation with complete white adipogenic medium including IBMX, troglitazone, dexamethasone, and insulin (Fig. 4e, f and Supplementary Fig. 5g). Moreover, stimulation with brown adipogenic

**Fig. 4 MME⁺ are a highly adipogenic fraction of PDGFRα⁺ FAPs. a, b** Representative images (**a**) and quantification (**b**) of proliferating MME⁻/⁺ FAPs during the first 48 h in culture, as measured by percentage of EdU⁺ nuclei (red, EdU⁺; white, Hoechst⁺, Scale Bar: 100 um). **c, d** Representative images (**c**) and quantification (**d**) proliferating MME⁻/⁺ FAPs during standard culture, as measured by percentage of EdU⁺ nuclei (red, EdU⁺; white, Hoechst⁺, Scale Bar: 100 um). **e, f** Representative images (**e**) and quantification (**f**) of MME⁻/⁺ FAP adipogenesis as measured by LD540⁺ area normalized to Hoechst⁺ area in full white adipogenic medium (red, LD540⁺; white, Hoechst⁺, Scale Bar: 100 um). **g, h** Representative images (**g**) and quantification (**h**) of MME⁻/⁺ FAP adipogenesis as measured by the number of adipocytes normalized to the number of cells (red, LD540⁺; white, Hoechst⁺, Scale Bar: 100 um) in low insulin medium. **i** Gene expression of *Pparg, Fabp4, Adipoq, Lep, Lpl, Cebpa, Plin1,* and *Mme* in MME⁻-derived and MME⁺-derived adipocytes after differentiation in low insulin medium. **j** Schematic representation of transplantation-mediated in vivo adipogenesis assay. **k, l** Representative images (**k**) and quantification (**l**) of mTomato(mT)⁺ adipocytes 14 days after transplantation of mT⁺MME⁻/⁺ FAPs into glycerol-injected tibialis anterior muscle as measured by the number of mT⁺ adipocytes per section at the mid-belly of the muscle. (green, perilipin⁺; gray, WGA⁺, fuchsia, mT⁺, Scale Bar: 100 um). **m** Representative images of adipocytes 14 days after transplantation of mT⁺MME⁻/⁺ FAPs into glycerol-injected tibialis anterior muscle at the mid belly of the muscle (green, perilipin⁺; white, Hoechst⁺, Scale Bar: 200 um). Each dot represents a single mouse. Bar graphs represent mean ± SEM. Student's *t* test (two-tailed, paired, *p < 0.05, ns > 0.05) was used in **b, f, h, l**. Student's *t* test (two tailed, unpaired) was used in **d**. Two way ANOVA with Sidak's multiple comparison test (*p < 0.05, **p < 0.01, ***p < 0.001) was used in **i**.

medium, including IBMX, rosiglitazone, dexamethasone, insulin, T3, and indomethacin, evoked a robust transcriptional response which was not different between MME⁺ and MME⁻ FAPs (Supplementary Fig. 5h). However, when we used minimal adipogenic induction medium, consisting of low glucose DMEM supplemented with 10% FBS and very low concentrations of insulin, we found that MME⁺ cells had a much higher capacity to differentiate into adipocytes than the MME⁻ fraction (Fig. 4g–i), suggesting that they are more sensitive to adipogenic differentiation. Interestingly, *Mme* expression was not different between the MME⁺ FAP derived and MME⁻ FAP derived adipocytes, suggesting that they reduce *Mme* during differentiation (Fig. 4i). Although MME⁺ FAPs are more highly adipogenic under minimal adipogenic induction, when isolated, brought in culture, and induced to differentiate into other mesenchymal lineages, they still possess differentiation capacity in the fibrogenic (*Acta2*) (Supplementary Fig. 5i), osteogenic (*Bglap2, Runx2*) (Supplementary Fig. 5j), and chondrogenic (*Sox9, Comp*) (Supplementary Fig. 5k) lineages.

To subsequently determine whether the MME⁺ FAPs contribute more readily to the development of adipocytes in vivo, we decided to perform a fate tracing experiment. We isolated MME⁺ and MME⁻ FAPs from PDGFRα-eGFP x Rosa^mTmG mice, which constitutively express the mTomato (mT) fluorescent protein and transplanted them into damaged WT muscle which had been injected with glycerol 3 days prior. This allowed us to track the fate of the injected mT⁺ cells (Fig. 4j). MME⁻mT⁺ and MME⁺mT⁺ FAPs isolated from one mouse were injected into the left or right leg of the same recipient mouse, respectively. Fourteen days after FAP injection, muscles were harvested to determine the contribution of the injected MME⁻mT⁺ and MME⁺mT⁺ FAPs to the development of adipocytes (perilipin⁺mT⁺) in the muscle. After transplantation into glycerol-injected muscle, MME⁻mT⁺ FAPs could be detected but rarely contributed to adipocytes. In striking contrast, we detected a significant contribution of MME⁺mT⁺ FAPs to adipocytes as determined by the number of mT⁺ adipocytes per muscle section (Fig. 4k, l). This was also reflected as an increase in the total perilipin⁺ area in MME⁺mT⁺ FAP-injected muscle as compared to MME⁻mT⁺ FAP-injected muscle (Fig. 4m and Supplementary Fig. 5l). These data confirm that MME⁺ FAPs are a subpopulation of FAPs that are characterized by high adipogenic potential.

**MME⁺ FAPs are characterized by reduced autocrine WNT/ GSK3 signaling and are refractory to Wnt-mediated inhibition of adipogenesis.** To gain further insight into the mechanisms which could lead to the increase in adipogenic potential of MME⁺ FAPs, we next performed bulk RNA sequencing on quiescent PDGFRa⁺(eGFP⁺) MME⁺ and PDGFRa⁺(eGFP⁺) MME⁻ FAPs which were freshly isolated from uninjured mouse

muscle. Principal component analysis showed that MME⁺ and MME⁻ FAPs clustered separately (Fig. 5a). Confirming our isolation protocol and in agreement with our scRNA-seq data, *Mme* was one of the most highly upregulated genes in our MME⁺ sorted samples (Fig. 5b). Additionally, the most differentially expressed genes in our bulk RNA-seq data were highly and specifically expressed in the mouse scRNA-seq dataset (Supplementary Fig. 6a) where all but three of the top differentially expressed genes were specifically expressed in the mouse scRNA-seq dataset (bold gene names, Fig. 5b). In agreement with the snRNA-seq and scRNA-seq data, *MME⁺* FAPs also showed upregulation of adipogenesis in a GSEA when compared to *MME⁻* FAPs (Fig. 5c, left panel). Furthermore, WNT ligand biogenesis and trafficking was downregulated in *MME⁺* FAPs (Fig. 5c, right panel). In agreement with those data, gene ontology (GO) enrichment analysis revealed that the top three downregulated molecular function terms in our MME⁺ cells were frizzled binding (known WNT receptors), receptor ligand activity, and signal receptor activator activity (Supplementary Fig. 6b). The gene lists associated with these terms each contain the canonical WNT ligands *Wnt2* and *Wnt10b*, along with other related genes such as *Rspo3* and *Dpp4* (Supplementary Fig. 6b). Among the WNT ligands, *Wnt2, 10b, 11, 9a,* and *1* were downregulated in MME⁺ FAPs while only *Wnt4* was slightly upregulated in MME⁺ FAPs (Supplementary Fig. 6c). Interestingly, *Wnt4* is known as a non-canonical WNT ligand with proadipogenic function[55]. We validated this downregulation of WNT signaling in our MME⁺ cells using qPCR analysis. We found the canonical WNT ligands *Wnt2* (Fig. 5d) and *Wnt10b* (Fig. 5e) to be lower in our MME⁺ population as compared to the MME⁻ population. This was also observable in our scRNA-seq datasets from mice (Supplementary Fig. 6d, e) and humans (Supplementary Fig. 6f,g). In freshly isolated FAPs from resting mouse muscle, decreased WNT signaling in the MME⁺ population coincided with an increase in the expression of *Cebpa*, whose expression in adipogenic progenitors is blocked by canonical WNT signaling[56], although no clear difference in *Pparg* was found (Supplementary Fig. 6h, i). To determine if this downregulation of canonical WNT ligand genes is associated with decreased intracellular WNT signaling through β-catenin, we examined the presence and localization of non-phosphorylated β-catenin via immunofluorescent staining. We found that active non-phosphorylated β-catenin levels were lower, especially in the nucleus, in freshly sorted MME⁺ cells (Fig. 5f, g) indicating decreased signaling through the canonical WNT pathway. After determining that canonical WNT signaling is lower in MME⁺ FAPs, we attempted to rescue their increased adipogenic capacity via pharmacologically activating the canonical WNT signaling pathway using CHIR99021, a GSK3β inhibitor[57], during adipogenic differentiation in minimal adipogenic medium. First, we

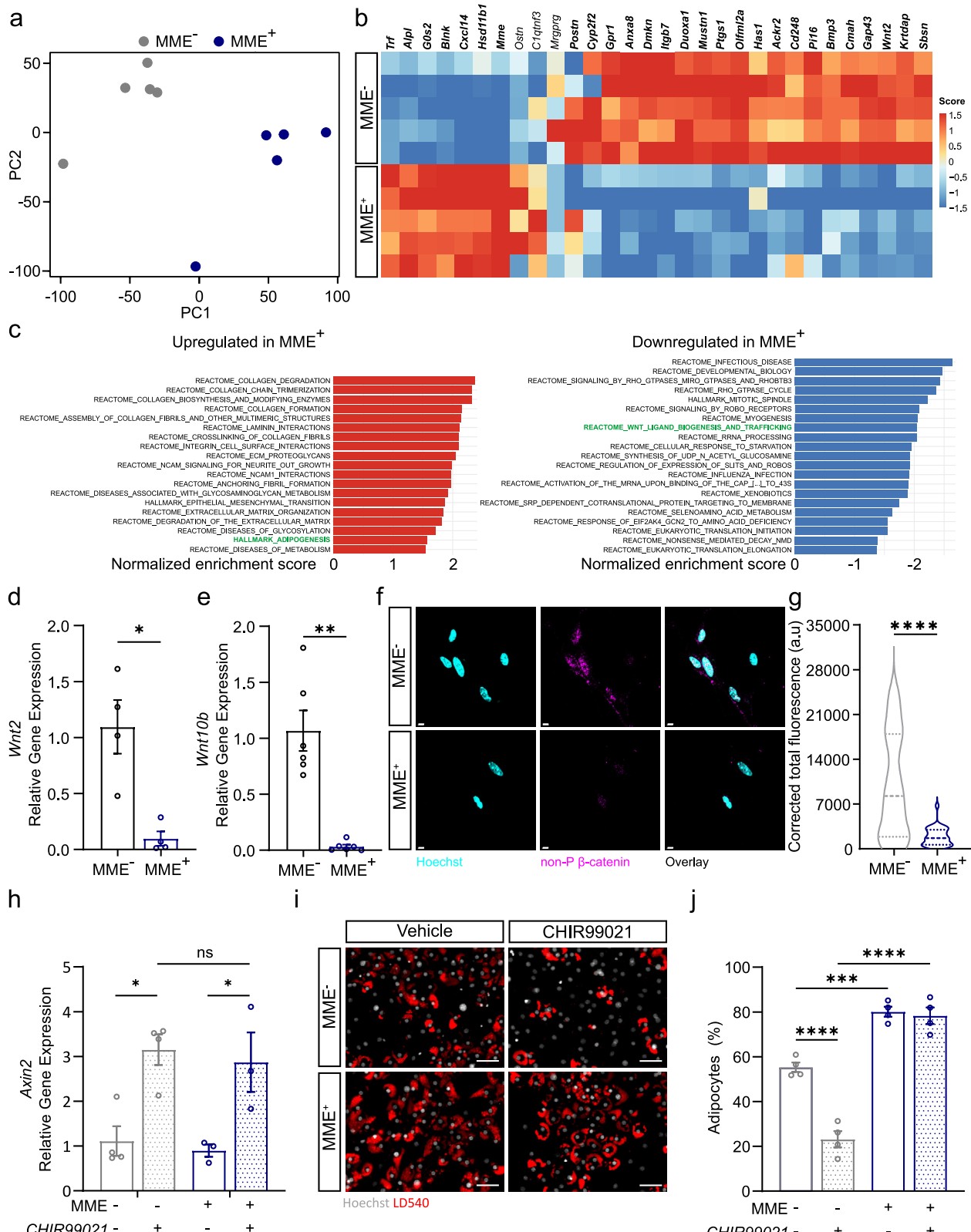

confirmed that CHIR99021 upregulated the expression of *Axin2* and *Nkd2* (Fig. 5h and Supplementary Fig. 6j), targets of canonical Wnt signaling activation[58]. Treatment with CHIR99021 inhibited adipogenesis in MME⁻ FAPs but had no inhibitory effect in MME⁺ FAPs (Fig. 5i, j). qPCR analysis of adipogenic genes confirmed these observations (Supplementary Fig. 6k, l). This indicates that MME⁺ FAPs are more prone to undergo adipogenesis and are refractory to the inhibitory function of WNT signaling.

**MME⁺ FAPs are more prone to apoptosis and are reduced after glycerol injection**. We next aimed to study the in vivo behavior of MME⁺ and MME⁻ FAPs in mouse muscle during

**Fig. 5 MME+ FAPs are characterized by reduced WNT signaling and refractory to WNT-mediated inhibition of adipogenesis. a** Principal Component Analysis (PCA) plot showing sample distances from the mouse bulk RNA-seq comparing *MME+* and *MME-* FAPs, color-coded by FAP subpopulation. **b** Heatmap with Z scores showing the top high variable genes identified in the mouse bulk RNA-seq comparing *MME+* and *MME-* FAPs. Each row represents and individual sample and each column a gene. Genes in bold were found to be differentially expressed in the mouse scRNA-seq (see Supplementary Fig. 6a). **c** Bar plots showing upregulated (left panel) or downregulated (right panel) pathways from GSEA of *MME+* vs *MME-* FAPs. X-axis indicates the normalized enrichment score (NES) for each pathway. Only significant pathways (adjusted $p < 0.05$) are shown. **d, e** Gene expression of *Wnt2* (**d**) and *Wnt10b* (**e**) in freshly isolated FAPs. **f, g** Canonical WNT signaling analysis: representative images (**f**) and quantification (**g**) of nuclear localized non-phosphorylated β-catenin as measured by corrected total fluorescence of non-phosphorylated β-catenin within a Hoechst+ nucleus (pink, non-P β-catenin+; blue, Hoechst+, Scale bar: 5 μm). **h** Relative gene expression of *Axin2* in MME−/+ FAP derived adipocytes after differentiation in low insulin medium with (CHIR99021) or without (Vehicle, DMSO) GSK-3 inhibition. **i, j** Representative images (**i**) and quantification (**j**) of MME−/+ FAP adipogenesis in low insulin medium with (CHIR99021) or without (Vehicle, DMSO) GSK-3 inhibition as measured by the number of adipocytes normalized to the number of cells (red, LD540+; white, Hoechst+, Scale bar: 50 μm). Each dot represents a single mouse. Bar graphs represent mean ± SEM. Student's *t* test (two tailed, unpaired, *$p < 0.05$, **$p < 0.01$, ***$p < 0.001$) was used in **d**, **e**, and **g**. Two-way ANOVA with Tukey's multiple comparisons test (***$p < 0.0001$) was used in **h** and **j**. Violin plots in **g** represent a representative experiment where at least 30 nuclei were quantified per condition.

regeneration. To do so, we used two injury models: intramuscular cardiotoxin (CTX) injection and intramuscular glycerol (GLY) injection, which have similar kinetics but different FAP fate[59]. Cardiotoxin injury leads to full regeneration whereas glycerol injury induces adipogenic differentiation leading to fatty infiltration. We injected cardiotoxin or glycerol into the tibialis anterior of 8–14-week-old mice and analyzed the behavior of MME+ FAPs. As previously described, muscle injury induces the activation and proliferation of FAPs which peaks 3 days post injury (dpi) leading to a rapid expansion of the number of FAPs[15,22,59]. In agreement with this, FAPs (eGFP+ cells) in both injury models incorporated EdU at 3 dpi but the fraction of proliferating FAPs was higher in glycerol injury than in cardiotoxin injury (Supplementary Fig. 7a, b). This was reflected as an increased number of FAPs at 3dpi after glycerol when compared to cardiotoxin injury (Supplementary Fig. 7c). Consistent with our in vitro results, both the MME+ and MME- FAP populations proliferated, but the fraction of proliferating MME+ FAPs was slightly higher than the MME- FAPs, mainly upon cardiotoxin injection (Fig. 6a, b). At this time point, the fraction of MME+ FAPs had increased well above baseline in both conditions (Fig. 6c, d). Because it has been shown that following the increase in FAPs at 3 dpi their numbers return to baseline with an accompanying wave of apoptosis[22], we subsequently evaluated MME+ FAP apoptosis upon muscle injury. To do so, we measured the fraction of cleaved caspase(c-caspase)3+ FAPs at 4 dpi, when the highest number of apoptotic FAPs is observed[22]. While the percentage of c-caspase3+ FAPs was not different between cardiotoxin injected and glycerol injected muscle at 4 dpi (Supplementary Fig. 7d, e), a much higher proportion of MME+ FAPs were c-caspase3+ as compared to MME- FAPs after both cardiotoxin and glycerol injury, indicating that muscle damage induces a selective apoptosis of MME+ FAPs (Fig. 6e, f). Finally, we found that, at 28 days after cardiotoxin injection when regeneration is completed, the percentage of MME+ FAPs normalized (Fig. 6g, h). However, 28 days post glycerol injection the percentage of MME+ FAPs fell below baseline (Fig. 6g, h). Since this reduction in MME+ fraction could not be explained by less proliferation nor increased cell death in the glycerol condition as compared to the cardiotoxin condition, it is likely the result of increased adipogenic differentiation of MME+ FAPs under adipogenic conditions (see above).

To further solidify our observations, we next repeated the transplantation experiment using mT+MME+ FAPs and mT+MME- FAPs which we injected in glycerol injured mice (for scheme see Fig. 4j). Twenty five days after the cell injection, we found fewer mT+ cells in the mT+MME+ FAP-injected leg than in the mT+MME- FAP-injected leg (Fig. 6i). Whether this is due to increased cell death, as would be expected due to their increased c-caspase3 expression after muscle damage (Fig. 6e, f),

or due to their differentiation into adipocytes, as would be expected due to their increased adipogenic capacity (Fig. 4g–m), is unknown. Furthermore, we found that a lower percent of the remaining mT+ cells were eGFP+ FAPs in the mT+MME+ FAP-injected leg than in the mT+MME- FAP-injected leg (Fig. 6j, k), suggesting that MME+ FAPs lose FAP characteristics under glycerol-injured conditions. In agreement with our pseudotime analysis (Supplementary Fig. 4f), we also observed that a significant fraction of MME- FAPs acquire MME (Fig. 6l, m), indicating that it could be possible for MME- FAPs to replenish the MME+ FAP population. To test this possibility, we performed a glycerol injury and waited 70 dpi to determine MME+ FAP abundance at this later timepoint after injury. Even though the data showed significant variability, we observed that MME+ FAP abundance normalized to baseline levels (Fig. 6n, o). Taken together, these data indicate that MME+ FAP differentiation into adipocytes after an acute injury leads to a transient reduction in their abundance which is later compensated by the progression of MME- FAPs into MME+ FAPs. We next wondered how the dynamic interplay between MME+ FAP adipogenic differentiation and the progression of MME- FAPs toward an MME+ phenotype would express itself in a chronic injury setting.

**Human MME+ FAPS are highly adipogenic and are reduced in fatty infiltrated human muscle.** To determine if MME+ FAPs are also reduced in humans under chronic degenerative adipogenic conditions, we decided to analyze muscle samples that differed in fatty infiltration levels. In this respect, it is known that patients with HOA have higher fatty infiltration in gluteal muscles (including *gluteus minimus*) when compared to other hip muscles[25,26]. To confirm those observations, we combined Magnetic Resonance Imaging (MRI) based quantification of hip skeletal muscle fatty infiltration with a thorough histological characterization of different hip muscles of end-stage HOA patients undergoing THA ($n = 12$). We confirmed higher fatty infiltration in *gluteus minimus* (GM) as compared to *rectus femoris* (RF) muscle using MRI (Fig. 7a, b). Moreover, H&E stainings on harvested muscle samples from these patients confirmed higher fatty infiltration in the GM (Fig. 7c and Supplementary Fig. 8a). To compare the cellular composition of a more highly fatty infiltrated muscle with a control muscle (with relatively low levels of fat infiltration), we characterized and compared a highly fatty infiltrated (GM^highFI) muscle and a control (RF^ctrl) muscle sample from the same patient of our existing dataset using scRNA-seq. Pre-surgery MRI-mediated quantification and post-surgery analysis of fatty infiltration confirmed that this patient had high levels of fibro-fatty infiltration in the GM^highFI muscle and lower levels in the RF^ctrl muscle (Fig. 7a, b and Supplementary Fig. 8a - blue dot). Expression values of each sample were independently log-normalized and then both

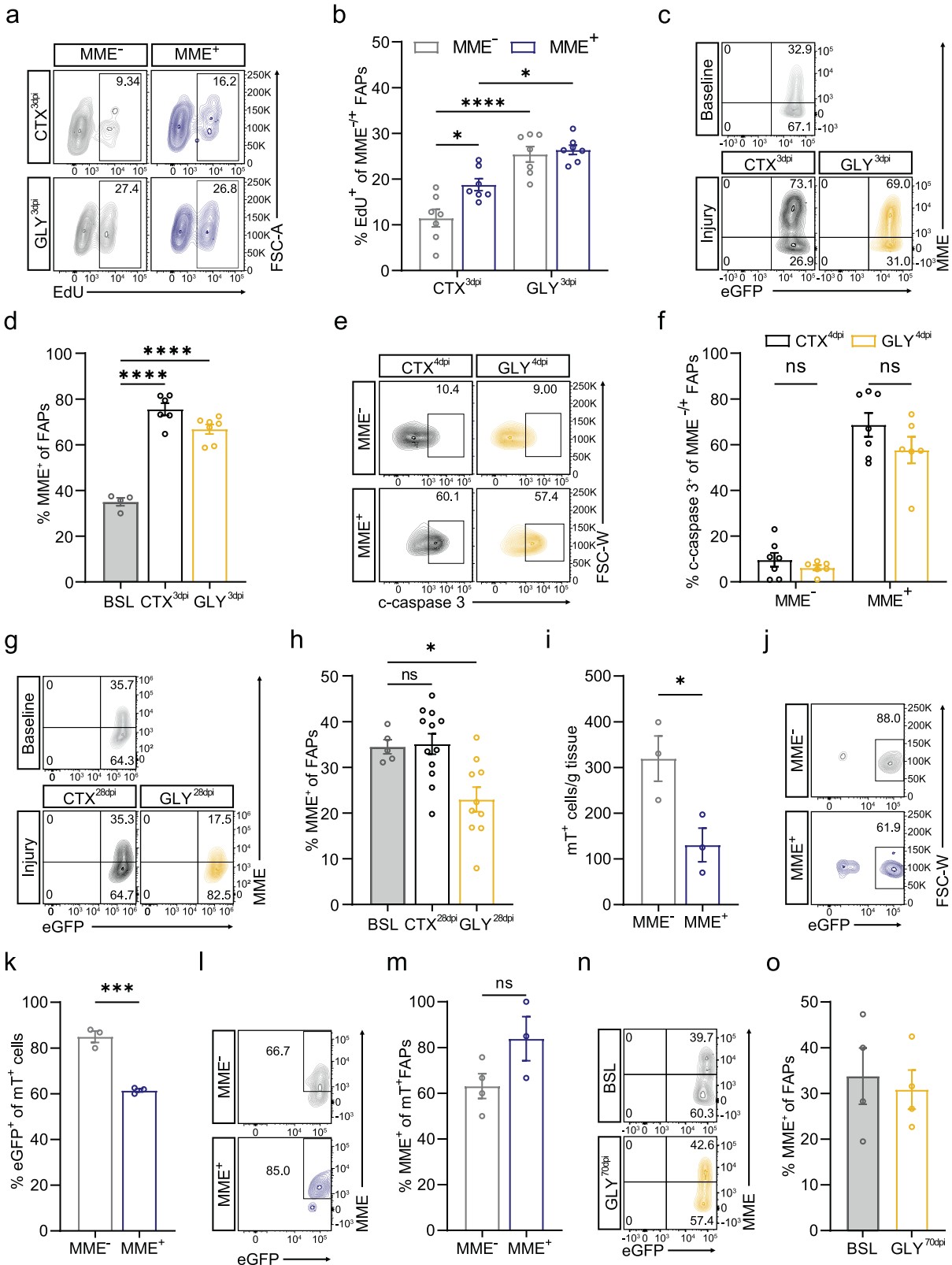

datasets were integrated and batch-effect corrected using Harmony[50]. We aimed at identifying which cell populations differed in the GM^highFI muscle as compared to the RF^ctrl muscle and found significant changes in the distribution of the different cell populations across each dataset. Along with a robust increase in the fractional abundance of immune cells in the GM^highFI muscle (Supplementary Fig. 8b), we found that the total number

of FAPs decreased from 58.03% from the total cell number in the RF^ctrl muscle to 27.88% in the GM^highFI muscle (Supplementary Fig. 8c). This decrease was almost exclusively accounted for by a massive reduction in the fractional abundance of *MME*+ FAPs while *CD55*+ and *GPC3*+ FAPS essentially retained their abundance (Supplementary Fig. 8b). Additionally, when analyzing the population redistributions within the FAP fraction only, we found

**Fig. 6 MME⁺ FAPS are more prone to apoptosis after injury and become reduced after glycerol injury. a**, **b** Representative flow cytometric analysis (**a**) and quantification (**b**) of EdU⁺ MME⁻/⁺ eGFP⁺ FAPs as a percentage MME⁻/⁺ FAPs at 3 dpi in CTX injected (CTX$^{3dpi}$) or GLY (GLY$^{3dpi}$) injected muscle. **c**, **d** Representative flow cytometric analysis (**c**) and quantification (**d**) of the MME⁺ FAP fraction under baseline (BSL) conditions as well as 3 dpi in CTX (CTX$^{3dpi}$) injected or GLY (GLY$^{3dpi}$) injected muscle. **e**, **f** Representative flow cytometric analysis (**e**) and quantification (**f**) of c-caspase3⁺ FAPs as a percentage of MME⁻/⁺ FAPs at 4 dpi in CTX injected (CTX$^{4dpi}$) or GLY (GLY$^{3dpi}$) injected muscle. **g**, **h** Representative flow cytometric analysis (**g**) and quantification (**h**) of the MME⁻/⁺ FAP fraction under baseline (BSL) conditions as well as 28 dpi in CTX injected (CTX$^{28dpi}$) or GLY injected (GLY$^{28dpi}$) muscle. **i** Quantification of the number of mT⁺ cells remaining in mT⁺MME⁻ FAP-injected muscle (MME⁻) or mT⁺MME⁺ FAP-injected muscle (MME⁺) at 28 dpi. **j**, **k** Representative flow cytometric analysis (**j**) and quantification (**k**) of the eGFP⁺ fraction of mT⁺ cells remaining in mT⁺MME⁻ FAP-injected muscle (MME⁻) or mT⁺MME⁺ FAP-injected muscle (MME⁺) at 28 dpi. **l**, **m** Representative flow cytometric analysis (**l**) and quantification (**m**) of the MME⁺ FAP fraction in mT⁺MME⁻ FAP-injected muscle (MME⁻) or mT⁺MME⁺ FAP-injected muscle (MME⁺) at 28 dpi. **n**, **o** Representative flow cytometric analysis (**n**) and quantification (**o**) of the MME⁺ FAP fraction at baseline (BSL) and in glycerol injected muscle at 70 dpi (GLY$^{70dpi}$). Each dot represents a single mouse. Bar graphs represent mean ± SEM. A two-way ANOVA with Tukey's multiple comparisons test (ns $p > 0.05$, *$p < 0.05$, ****$p < 0.0001$) was used in **b** and **f**. A one-way ANOVA with Holm-Sidak's multiple comparison test (*$p < 0.05$, ****$p < 0.0001$) was used in **d** and **h**. Student's $t$ test (two tailed, unpaired, *$p < 0.05$, **$p < 0.01$) was used in **i**, **k**, **m**, and **o**.

that *MME⁺* cells went from 49.54% of the total FAPs to a 26.09% in the GM$^{highFI}$ muscle while the *CD55⁺* fraction increased from 35.97% to 54.35% (Fig. 7d and Supplementary Fig. 8d). Finally, we used immunofluorescence staining to confirm the reduction of the fractional abundance of *MME⁺* FAPs in the GM$^{highFI}$ muscle in the same patient population ($n = 12$) and observed a lower MME⁺ area in the GM$^{highFI}$ muscle when compared to the RF$^{ctrl}$ muscle within the same patient (Fig. 7e, f). We measured lower MME⁺ area in GM muscle in 10 out 12 patients (Fig. 7f). Thus, *MME⁺* FAPs are reduced in FI muscle tissue.

Ultimately, to show that human MME⁺ FAPs are also more highly adipogenic than their MME⁻ counterpart, we isolated MME⁺ and MME⁻ FAPs from human muscle. FAPs were isolated as (MME⁺ or MME⁻)CD34⁺CD31⁻CD45⁻CD56⁻PI⁻ via FACS (Supplementary Fig. 8e) and brought in culture as described before[1]. When reaching 95% confluence, the medium was changed to an adipogenic differentiation medium. In agreement with our mouse data, human MME⁺ FAPs differentiated much more readily into adipocytes than the MME⁻ FAPs (Fig. 7g, h). Interestingly, the differences in adipogenic potential of human MME⁺ and MME⁻ FAPs were apparent under full adipogenic stimulation where we found very few MME⁻ FAPs undergoing adipogenic differentiation. Together, these findings show that, in both mice and humans, MME⁺ FAPs are a highly adipogenic FAP fraction and that MME⁺ FAPs are reduced under adipogenic conditions.

## Discussion

In this study, we used scRNA-seq approaches to characterize FAP heterogeneity in human skeletal muscle. Given their ability to differentiate into adipocytes, we compared these FAP subtypes with known adipose tissue derived progenitor subtypes. We found that both adipose tissue and muscle share a common interstitial progenitor subtype (CD55⁺ FAPs resemble DPP4⁺ adipogenic progenitors), but we also report the existence of unique FAP lineages in muscle (GPC3⁺ and MME⁺) which do not share their transcriptional signature with reported pre-adipogenic (ICAM⁺ or CD142⁺) cells. We identified MME⁺ FAPs as a subpopulation of FAPs which is reduced in fatty infiltrated muscle and used the membrane presence of MME to isolate, culture and characterize MME⁺ FAPs. MME⁺ FAPs exhibited remarkable propensity for adipogenic differentiation ex vivo as well as in transplantation experiments and become reduced after chronic adipogenic injury in humans.

Only a limited number of studies have thus far investigated FAP heterogeneity at a single cell level in humans[1,23,24]. Initial studies evaluated the cellular composition of the FAP population in muscle of young healthy volunteers[23,24]. The transcriptional properties of those datasets nicely overlapped with our

observations, not only showing the validity of our approach but also suggesting that the transcriptional properties of FAPs in healthy human muscle remain relatively stable during aging. This is likely different during a pathological setting: in skeletal muscle of obese type 2 diabetes patients, Farup et al.[1] reported the enrichment of THY1/CD90⁺ FAPs which are poised for ECM production during fibrotic settings. Our patient population was mainly composed of non-obese (BMI < 35), active, and healthy people with HOA, who are characterized by the selective fatty infiltration of some muscles surrounding the affected joint. We did not pick up significant THY1 expression in our single cell dataset, and COL1A1 levels (characteristic of THY1/CD90⁺ FAPs) were particularly low in our MME⁺ population when compared to the CD55⁺ and GPC3⁺ populations. If THY1/CD90⁺ FAPs are at all present within our patient population, they are thus likely to be part of the MME⁻ fraction. In the fatty infiltrated muscle, we observed a strong reduction in the number of MME⁺ FAPs when compared to non-infiltrated muscle of the same joint and within the same patient. These data suggest that specific pathological settings differentially affect FAP heterogeneity and composition, likely reflecting their specific contribution to fibro- or fatty-degeneration in vivo.

We found that MME⁺ FAPs isolated from mouse and human muscle were more adipogenic than the MME⁻ FAPs. This difference was only observed under low adipogenic conditions when we used murine FAPs but was even observed while using full adipogenic medium in human FAPs. In the latter setting, only very few MME⁻ FAPs underwent adipogenic differentiation. Besides illustrating the differences between models (mouse versus human) and raising caution concerning the interpretation of data using full adipogenic medium in mouse FAP studies, these data highlight the functional relevance of MME⁺ FAPs as a dominant adipogenic FAP population in skeletal muscle. To the best of our knowledge, we did not find another report which shows different adipogenic potential between different human FAP populations. We observed that MME⁺ FAPs have very low expression of canonical WNT ligands, as well as decreased canonical WNT signaling through β-catenin. Canonical WNT signaling is a potent negative regulator of adipogenesis[56] and keeps adipogenic progenitors in an undifferentiated state through the inhibition of the expression of adipogenic transcription factors such as *Cebpa* and *Pparg*[60]. WNT signaling has previously been shown to control adipogenic differentiation of FAPs in a β-catenin dependent manner[61]. While Reggio et al. focused on the ubiquitously expressed Wnt5a, their work illustrated the differential expression of Wnt2 and Wnt10b between FAP subpopulations, similar to our observations[61]. We found that β-catenin indeed regulates FAP adipogenesis but only in those cells governed by autocrine WNT signaling, such as the MME⁻ FAPs. In contrast, MME⁺ FAPs, which do not express canonical WNT ligands nor have

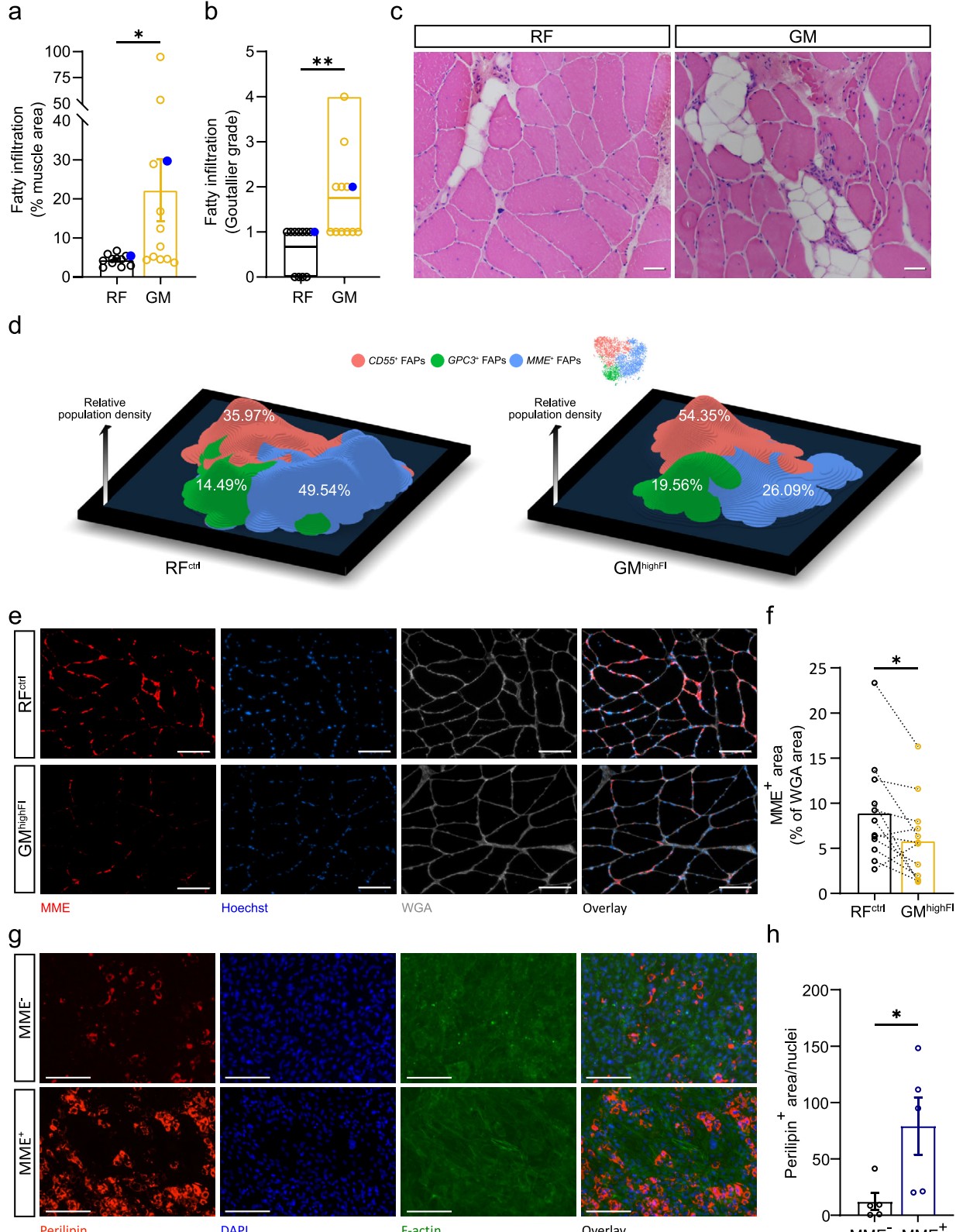

14      COMMUNICATIONS BIOLOGY | (2023)6:111 | https://doi.org/10.1038/s42003-023-04504-y | www.nature.com/commsbio

appreciable levels of non-phosphorylated β-catenin, remained highly sensitive to the adipogenic effect of insulin even after the pharmacological inhibition of GSK3β. This lack of sensitivity to the anti-adipogenic effects of canonical WNT activation through inhibition of GSK3β was independent of their ability to transcriptionally respond through the upregulation of downstream targets such as *Nkd2* and *Axin2*. Considering these data, it remains to be determined if/how, and to which extent, the regulation of canonical WNT signaling in FAPs in vivo controls the balance between complete regeneration and fatty degeneration. Furthermore, the exact molecular mechanism governing the enhanced adipogenic capacity of MME+ FAPs, potentially

**Fig. 7 Human MME+ FAPs are highly adipogenic and are reduced in fatty infiltrated human muscle. a, b** Relative fatty infiltrated area (**a**) and Goutallier grade (**b**) assessed in MRI images of the RF and GM muscles. Biopsies from the patient used for scRNA-seq are marked with blue dots. **c** Representative images of haematoxylin and eosin (H&E)-stained sections of the RF and GM. **d** 3D TSNE density plots showing the abundance of FAP subpopulations in the RF^ctrl (left panel) and GM^highFI (right panel) scRNA-seq datasets. Height of the different areas defines the abundance of FAP subpopulations. Color-coded by the identified subpopulations. **e, f** Representative images (**e**) and quantification (**f**) of MME+ area normalized to WGA+ area in RF^ctrl and GM^highFI muscles (Scale bar: 100 um). **g, h** Representative images (**g**) and quantification (**h**) of MME−/+ FAP adipogenesis as measured by perilipin+ area normalized to number of nuclei (red, perilipin+; blue, DAPI+; green F-actin+; Scale bar: 150 um). Each dot represents individual patients. Bar graphs represent mean ± SEM in **a**, **f**, and **h**. Student's *t* test (two tailed, ratio-paired, *$p < 0.05$) was used in **a**, **f**, and **h**. Student's *t* test (two tailed, paired, **$p < 0.01$) was used in **b**. Box plot in **b** represents the minimum, maximum, and median.

through interacting with the tissue microenvironment, should still be investigated.

The observation that MME+ FAPs are highly prone to undergo adipogenic differentiation, also prompted us to study the signals that prevent this differentiation during normal conditions. Recent literature has shown that upon muscle injury, the number of FAPs is heavily fluctuating: shortly after injury, an initial burst of FAP proliferation is observed which is followed by FAP apoptosis. Both FAP proliferation and apoptosis are highly influenced by the inflammatory state of the muscle, in particular the balance between the pro- or anti-inflammatory phenotype of macrophages inside the muscle[22]. Upon muscle damage, circulating monocytes infiltrate the muscle and rapidly differentiate into pro-inflammatory macrophages. In *Ccr2*−/− mice for instance, FAPs do not undergo apoptosis due to lack of infiltrating pro-inflammatory macrophages which secrete TNFa. During normal regeneration, these pro-inflammatory macrophages however rapidly repolarize into pro-regenerative M2-like macrophages which promote the deposition of the ECM through the secretion of TGFβ[22]. Moreover, IL-4 secreted by eosinophils promotes FAPs proliferation and inhibits their adipogenic differentiation[21]. However, whether specific FAP populations are differentially affected during normal versus aberrant (adipogenic) conditions, has not been described so far. We observed that almost 70% of MME+ FAPs were c-caspase3+ during muscle regeneration following cardiotoxin as well as glycerol injection, suggesting that those cells preferentially undergo apoptosis and are mostly cleared before they can undergo adipogenic differentiation. In contrast, only very few MME- FAPs were c-caspase3+. Also, after glycerol injury the proportion of MME+ FAPs fell below baseline levels at 28 dpi, likely because they differentiate into adipocytes leading to their reduction at this time point after glycerol injury. At this moment, it is unclear which signals prevent the adipogenic conversion of MME+ FAPs after cardiotoxin-induced injury. Recent literature evidence in adipose tissue[49] as well as muscle[62] has shown the presence of an F3/CD142 population which inhibits the maturation and adipogenic differentiation of early progenitors. Moreover, it was observed that there is a deficiency of this population under dystrophic conditions which is proportional to increased adipogenesis[62]. However, similar to recent work[52], we detected more widespread expression of CD142/F3 in our dataset in all populations. All transcriptomic data generated in this study can be fully explored at https://shiny.debocklab.hest.ethz.ch/Fitzgerald-et-al/. If anything, CD142/F3 was slightly higher in our MME+ FAPs. Further research using more specific markers should thus evaluate whether the inhibitory population specifically controls the fate of MME+ FAPs.

Both by using glycerol injections, a well described preclinical model of muscle fatty infiltration, as well as by comparing paired human muscle samples with different levels of fatty infiltration, we found that MME+ FAPs become reduced under adipogenic conditions. By twenty-eight days after glycerol injection, the number of MME+ FAPs had not recovered to baseline levels. Since transplantation experiments showed that more MME+ FAPs differentiated into adipocytes, their

pronounced sensitivity to differentiate likely explains why less MME+ FAPs were present at this time point after an acute injury, however we acknowledge the possibility for other mechanisms operating that could explain this phenomenon. It took until 70 days after glycerol injection for the levels of MME+ FAPs to recover to baseline levels. This apparent reduction and later recovery of the MME+ FAP fraction is likely due to their differentiation into adipocytes outpacing their replenishment by the MME- FAPs, which we observed could become MME+ after transplantation into glycerol injected muscle. Moreover, we found that in the chronic damage condition which results from HOA there were fewer MME+ FAPs in fatty infiltrated *gluteus minimus* versus *rectus femoris* both in the scRNA-seq experiment as well as in the histological validation. The reasons for this chronic reduction in MME+ FAPs remain outstanding. We speculate that, in this chronic injury situation, the reduction in MME+ FAP levels could be due to their differentiation rate exceeding their rate of replenishment. Whether this observation implies that there is a limited capacity for fatty infiltration in muscle under specific conditions remains to be determined.

Taken together, we have identified an MME+ FAP subpopulation with a pronounced adipogenic potential ex vivo and in vivo. Understanding why MME+ FAPs are more prone to adipogenic differentiation and which signals allow this differentiation to occur, may provide opportunities to therapeutically target specific FAP populations to prevent fatty degeneration.

**Limitations of the study**. While we have demonstrated a cell autonomous increase in adipogenic capacity in MME+ FAPs, we have not addressed whether and how their adipogenic differentiation is controlled by the microenvironment, a crucial mediator of FAP fate specification[16]. In particular, the appearance of vascular markers expressed by MME+ FAPs suggests that those might interact with adjacent blood vessels. Furthermore, it would be insightful to employ the use of genetic lineage tracing models to definitively show the increased adipogenic capacity of MME+ FAPs in vivo.

Additionally, we have assessed the capacity of both MME+/− FAPs to differentiate in vitro into the fibrogenic, osteogenic and chondrogenic lineages. However, whether these FAP populations also keep this potential in vivo remains to be elucidated.

## Methods

**Patients and study design**. For histological, single cell, and single nuclear transcriptomic analysis comparing muscles of different levels of fatty infiltration, twelve patients undergoing total hip arthroplasty (THA) for symptomatic hip osteoarthritis (HOA) were recruited from the Department of Hip Orthopaedic Surgery, Schulthess Clinic (Zurich, Switzerland). The inclusion criteria were end-grade HOA based on clinical and radiological examinations, scheduled for THA, men, and age between 55 and 75 years. Exclusion criteria were pain on the contralateral hip, lower limb surgeries in the previous 10 years, inability to walk without aids, BMI > 35 kg*m−2, and cardiorespiratory diseases. All patients (67 ± 6 yrs, 84 ± 10 kg, 179 ± 4 cm) signed an informed consent before participating in the study. Prior to surgery, the degree of fatty infiltration in the gluteus minimus (GM) and rectus femoris (RF) was assessed using MRI. During the extracapsular approach for THA, the GM and RF muscles were exposed and biopsies were

collected for sequencing and histological analysis. The study was conducted according to the Declaration of Helsinki and the protocol was approved by the Ethics Committee of the Canton of Zurich (KEK number: 2016-01852).

**Mice.** PDGFRa-eGFP mice, which express the H2B-eGFP fusion gene from the endogenous *Pdgfra* locus, were used to isolate FAPs from the muscle[63]. PDGFRa-eGFP mice were bred with Rosa26[mTmG] mice[64] in order to isolate FAPs and trace their fate after transplantation into wild type mice. All animals used in this study were of C57BL/6 background. A mix of male and female animals aged between 8–14 weeks at the start of the experiment were used. Mice were randomly allocated to different treatment groups, and the investigator was blinded to the group allocation during the experiment as well as during the analysis. All mice were housed at standard housing conditions (22 °C, 12 h light/dark cycle), with ad libitum access to chow diet (18% proteins, 4.5% fibers, 4.5% fat, 6.3% ashes, Provimi Kliba SA) and water. Health status of all mouse lines was regularly monitored according to FELASA guidelines. All animal experiments were approved by the local animal ethics committee (Kantonales Veterinärsamt Zürich, licenses ZH180/18 and ZH094/17), and performed according to local guidelines (TschV, Zurich) and the Swiss animal protection law (TschG).

Cardiotoxin injury was used to model regeneration which completely resolves and does not lead to lasting fatty infiltration. Glycerol injury was used to model fatty degeneration where muscle regeneration is incomplete and leads to lasting fatty infiltration. Mice were anesthetized with isoflurane, the hind limb was shaved and sterilized, and 50 μL of either 10 μM cardiotoxin diluted in PBS or 50% v/v glycerol in HBSS was injected into the tibialis anterior muscle while slowly removing the needle in a proximal to distal fashion.

**Assessment of human muscle fatty infiltration.** The degree of fatty infiltration in the GM and RF muscles were assessed by MRI without contrast using a 1.5-T system (Avanto-fit, Siemens Healthcare, Erlangen, Germany) according to a standardized protocol. A combination of an 18-channel surface coil and a 32-channel spine coil was used. Coronal T2-weighted fast spin-echo (3990/59 [repetition time msec/echo time msec], echo train length of 15, 4-mm section thickness, 220 × 220-mm field of view, 512 × 512 matrix), sagittal T1-weighted spin-echo (641/10, echo train length of 3, 3-mm section thickness, 210 × 210-mm field of view, 320 × 320 matrix), transverse short inversion time inversion-recovery (5550/36/150 [repetition time msec/echo time msec/inversion time msec], echo train length of 13, 7-mm section thickness, 180 × 180-mm field of view, 320 × 320 matrix), transverse T1-weighted spin-echo (434/10, echo train length of 2, 6-mm section thickness, 200 × 200-mm field of view, 512 × 512 matrix), and transverse Dixon (6.7/2.4, 5-mm section thickness, echo train length of 2, 200 × 200-mm field of view, 160 × 160 matrix) MR images were obtained. For quantitative assessment of fatty infiltration, fat signal fraction maps were generated from the Dixon sequence[65]. For both the qualitative and the quantitative assessments the GM muscle was assessed on transverse T1-weighted MR images and the fat signal maps, respectively, at the level of the upper border of the acetabular rim[66], the RF muscle at the level of femoral head center[67]. The degree of fatty infiltration was evaluated quantitatively with the fat signal fraction maps (0–100%), and qualitatively using the Goutallier grading scale ranging from 0 to 4 as follows: grade 0, no intramuscular fat; grade 1, some fat streaks; grade 2, fat is evident but less fat than muscle tissue; grade 3, equal amounts of fat and muscle tissue; grade 4, more fat than muscle tissue[9].

**Human muscle biopsy collection and analysis.** GM and RF biopsies were collected from all patients by the same hip surgeon (ML) during the extracapsular part of the direct anterior approach used to perform THA. Depending on the patient, a biopsy sample was collected from each muscle using a scalpel and was immediately processed for scRNA-seq or was snap frozen in liquid nitrogen for subsequent snRNA-seq analysis. For single cell analysis, samples were placed in ice cold PBS and transferred to the laboratory within 30 min. From all patients, another piece of muscle sample was rapidly embedded in Optimal Cutting Temperature (OCT) medium (6478.1, Carl Roth), frozen in liquid nitrogen-cooled isopentane and stored at −80 °C.

**Human ex vivo FAP adipogenic differentiation.** Human skeletal muscle samples were immediately submerged into a 4 °C wash-buffer [Ham's F10 incl. glutamine and bicarbonate (Cat no. N6908 Sigma, Sigma-Aldrich, Denmark) 10% Horse serum (cat no. 26050088, Gibco, Thermo Fisher Scientific, MA, USA), 1% Penstrep (cat no 15140122, Gibco)]. Muscle was transported from the operation suite to the laboratory within 15 min in ice-cold wash-buffer. Upon arrival the muscle biopsy was initially dissected free of visible tendon/connective tissue and fat. The biopsy was then divided into pieces of up to 0.5–0.8 grams and briefly mechanically minced with sterile scissors. The muscle slurry was then transferred to C-tubes (130-093-237, Miltenyi Biotec) containing 8 ml wash-buffer including 700 U/ml Collagenase II (lot 46D16552, Worthington, Lakewood, NJ, USA) and 3.27 U/ml Dispase II (04 942 078 001, Roche Diagnostics). Mechanical and enzymatic muscle digestion was then performed at 37 °C on the gentleMACS with heaters (130-096-427, Miltenyi Biotec) for 60 min using a skeletal muscle digestion program (37C_mr_SMDK1). When digestion was

complete 8 ml wash buffer was added to the single cell solution and this was filtered through a 70 μm cell strainer and washed twice to collect any remaining cells. The suspension was centrifuged at 500 g for 5 min and the supernatant removed. The cell pellet was resuspended in freezing buffer (StemMACS, 130-109-558, Miltenyi Biotec) and stored 1–3 weeks at −80 °C. Approximately 1.5 h before FACS the frozen cell suspension was thawed until a small amount of ice was left and resuspended in 10 ml wash-buffer. The solution was centrifuged at 500 g for 5 min and the supernatant removed to clear the freezing buffer. The cells were then resuspended in wash buffer and incubated in MACS human FcR blocking solution (20 μl/sample, 130-059-901, Miltenyi Biotec) and primary antibodies against CD45-FITC (12 μl/sample, 130-114-567, Clone 5B1, Miltenyi Biotec), CD31-FITC (4 μl/sample, 130-110-668, Clone REA730, Miltenyi Biotec), MME-PE (4 μl/sample, Clone REA877, 130-114-693, Miltenyi Biotec), CD56-BV421 (1:100, 562752, BD Bioscience), CD82-PE-vio770 (130-101-302 10 μl/sample, Miltenyi Biotec) and CD34-APC (20 μl/sample, clone 581, BD Bioscience) in darkness at 4 °C for 30 min. Propidium iodide (PI, 10 μl/sample, 556463, BD Bioscience) immediately before sorting to exclude non-viable cells. The suspension was washed in 10 ml wash-buffer and centrifuged at 500 g. Finally, the sample resuspended in wash-buffer and filtered through a 30 μm filter to remove any remaining debris/aggregates. Non-stained cells and single-color controls were prepared in combination with the primary (full color) samples. To ensure bright single-color controls for compensation, compensation beads (01-2222-41, eBioscience, Thermo Fisher Scientific) was utilized. Cell sorting was performed using a FACS-AriaIII cell sorter (BD Bioscience) with 405 nm, 488 nm, 561 nm and 633 nm lasers. A 100 μm nozzle at 20 psi was utilized to lower pressure/stress on the cells as well as prevent clogging. Gating strategies were optimized through multiple earlier experiments, which included various full color samples, unstained sample, single color samples and fluorescence minus one (FMO) controls for CD45, CD31, CD34, MME, CD82 and CD56. Cells were sorted into 4 °C cooled collection tubes containing wash-buffer. Data was collected in FACSdiva software and later analyzed in FlowJo (FlowJo 10.6.1, BD). The following populations were FACS isolated: CD34+MME-CD56-CD45-CD31-PI- (MME+ FAPs), CD34+MME+CD56-CD45-CD31-PI- (MME+ FAPs). The purity of the populations was checked following the sort by re-running the samples.

All human cell culture experiments were performed without passages to limit cell-culture artifacts and at 37 °C and 5% CO₂. All tissue-culture plates or chamberslides were treated with extra-cellular matrix (ECM gel, E1270, Sigma) and chamberslides were in addition pre-treated with Poly-D-Lysine (A-003-E, Millipore, Sigma) to increase adherence of the ECM. Cells from FACS were initially plated at a density of approximately $1 \times 10^4$ cells/cm². All cells were plated in wash-buffer and after 24 h this was switched to a growth media (GM, DMEM (4.5 g/L glucose) + 20% FBS (16000044, Thermo Fisher Scientific) + 10% Horse serum + 5 ng freshly added bFGF (human rbFGF, F0291, Sigma)). Media was changed every 2–3 days. When reaching >95% confluency, the media was switched to a differentiation media. For adipogenic differentiation we used a complete adipogenic differentiation media (ADM, 130-091-677, Miltenyi Biotec) for 12 days and replaced media every other day.

For cells in plates or chamberslides, these were fixed for 10 min in 4% paraformaldehyde, washed twice in PBS and stored in PBS until staining. Cells were then incubated for 30 min in blocking buffer (10% goat serum, 0.2% Triton-100X in PBS). Afterwards cells were incubated with primary antibody against Perilipin-1 (rabbit, 1:200, Cell Signaling Technologies) and alpha-smooth muscle actin (1:150, A5228, Sigma) at 4 °C overnight in blocking buffer. This was followed by incubation with the secondary antibody Alexa-fluor 647 goat-anti-mouse, Alexa-fluor 568 goat-anti-rabbit (1:500; A-21235 and A-11011, Invitrogen, Thermo Fisher Scientific) and Alexa-fluor 488 Actingreen (F-actin/Phalloidin) ready probe (R-37110, Invitrogen, Thermo Fisher Scientific) for 1.5 h at room temperature. Finally, cells were washed 3 × 5 min, with one wash containing DAPI, and either maintained in PBS or mounted in mounted media. Minus primary controls were included for all stains during optimization to ensure specificity. Imaging was performed. Images were acquired using an EVOS M7000 automated imaging system (Thermo Fisher Scientific). ImageJ was utilized for threshold and quantification of percentage positive area/foci.

**Human cell isolation for single cell sequencing.** Muscles were minced with scissors until a homogeneous paste-like mash was formed. Thereafter, the mashed muscle was enzymatically digested in digestion buffer containing 2 mg/ml Dispase II (D4693, Sigma-Aldrich, Steinheim, Germany), 2 mg/ml Collagenase IV (17104019, Thermo Fisher Scientific, Zurich, Switzerland), 2 mM CaCl2 and 2% BSA in PBS for 30 min at 37 °C with gentle shaking every 5 min. The reaction was stopped by adding an equal volume of TC media containing low glucose (1 g/L) DMEM supplemented with 10% FBS (10270-106, ThermoFisher Scientific) and the suspension was passed through a 100-μm cell strainer (#431751, Corning, New York, USA). Subsequently the suspension was centrifuged at 350 g and the supernatant discarded. The pellet was then suspended in 2 mL of erythrocyte lysis buffer (154 mM NH₄Cl, 10 mM KHCO₃, 0.1 mM EDTA, pH = 7.35) and incubated on ice for approximately 1 min (longer for samples with more blood). 20 mL of cold PBS was then added to the suspension which was then immediately filtered through a 40-μm cell strainer (431750, Corning, New York, USA) to remove tissue

debris. Cell suspension was centrifuged at 550 g for 5 min at 4 °C. The pellet was then suspended in PBS/BSA (0.5% BSA) and brought for FACS. Calcein Violet (65-0854-39, ThermoFisher Scientific) and 7AAD (A1310, ThermoFisher Scientific) were added directly before sorting.

**Isolation of human nuclei.** For snRNA-seq, nuclei isolation protocol was adapted from ref. [28]. Briefly, human muscle biopsies were collected and immediately snap frozen in liquid-N$_2$ and stored at −80 °C until further processing. Samples were thawed on ice for 5 min on ice in a petri dish before adding a few drops of ice-cold lysis buffer (NaCl 146 mM, Tris-HCl 10 mM, CaCl$_2$ 1 mM, MgCl$_2$ 21 mM, 0.5% CHAPS, 0.1% BSA, SUPERase• In™ RNase Inhibitor 20 U/mL (AM2696, ThermoFisher Scientific). Muscles were minced with a scalpel until a paste-like mash was formed. The minced tissue was added to a 15 mL Dounce homogenizer with 5 mL of ice-cold lysis buffer and allowed to lyse for an additional 3 min on ice. 9 mL of wash buffer (2% BSA in PBS with 20 U/mL SUPERase• In™ RNase Inhibitor) was then added and the suspension and pestle A was lowered/raised 10 times. The suspension was then serial filtered through a 70 μm and 40 μm strainer and transferred to a new 15 mL conical tube. The suspension was then centrifuged at 500 g and the supernatant removed via pipet. The pellet was then washed with 10 mL wash buffer and centrifuged at 500 g. The supernatant was removed, and the pellet transferred to a 5 mL round bottom FACS tube. The sample was sorted on a BD FACSAria Fusion without cooling the sample holder to avoid coagulation of any remaining fats in the sample. Hoechst$^+$ nuclei were sorted into wash buffer and brought immediately for loading into the 10X Chromium instrument.

## Transcriptomic analysis

*Human snRNA-seq analysis.* Human muscle-derived nuclei were FACS sorted based on Hoechst$^+$ staining and loaded in the 10X Chromium Next GEM Chip G, targeting the recovery of ~10000 cells. During nuclei isolation, different patients ($n = 5$) were pooled together in two groups: group1 ($n = 4$, accession codes: GSM6034993 and GSM6034994) and group2 ($n = 5$, accession codes: GSM6034995 and GSM6034996). 4 of the group2 muscles had a paired group1 muscle from the same patient, the remaining group2 muscle did not have a paired group1 muscle. Then each muscle group, was loaded in 2 wells (total of 4) in the 10X Chip. Libraries were generated following 10X Chromium Next GEM Single Cell 3' Reagent Kits v3.1 (CG000204 Rev D) and sequenced on the Illumina Novaseq 6000 system. Sequencing reads were aligned using Cell-Ranger 6.0.2 and the GENCODE GRCh38.p13 annotation release 37. The command-line–*include introns* was used to also count reads mapping to intronic regions. Each individual sample was processed independently for quality control (QC) and normalization steps using *scran 1.22.0 and scater 1.22.0* packages[68,69] in R (version 4.1.2). During QC steps, high quality nuclei were selected based on three parameters: library size, number of detected genes and percentage of reads mapped to mitochondrial genes. Nuclei with more than 3 median absolute deviation (MAD) from the median value of any of the previous parameters were discarded for downstream analyses. Counts were log-normalized based on library size factors. After individual QC and normalization, samples were pooled and batch-corrected using Harmony[50]. Dimensionality reduction by t-Distributed Stochastic Neighbor Embedding (TSNE) was calculated based on the dimensions generated by Harmony. Clustering was performed by building a nearest-neighbor graph ($k = 50$) and then using the Louvain algorithm for community detection. We used *Seurat* package (version 4.0.5)[70] for marker gene detection using a threshold of 0.25 for minimum log fold change and fraction of detection. After clustering and marker gene detection, FAPs and adipocytes were selected for downstream analyses based on the differential expression of known FAPs (PDGFRA, DCN, CD34) and adipocytes (PLIN1, PPARG, ADIPOQ) markers. For GSEA, we used *fgsea* package (version 1.20.0)[71] and the Hallmark pathways from MSigDB[30]. Differentially expressed genes for GSEA were calculated using a negative binomial distribution from the DESeq2 method within the Seurat package[70]. For GSVA, we used *GSVA* package (version 1.44.2)[31] and the "Adipogenesis" pathways from Hallmark (MsigDB)[30] and WikiPathways[32]. After calculation of a GSVA score per cell for each pathway, we then calculated a GSVA Z-score per cell for each pathway using the following formula: (cell GSVA score – average GSVA score) / standard deviation GSVA score. For demultiplexing the different donors in the full snRNA-seq data, we first used *cellsnp-lite* package (version 1.2.2)[72] for SNP genotyping in the bam files using a reference preprocessed SNP list from 1000 Genomes Project Consortium[73]: genome1K.phase3.SNP_AF5e2.chr1toX.hg38. Then, we applied *vireo* package (version 0.5.7)[29] following the manual provided by the authors. As the pooled ($n = 5$) snRNA-seq data was generated in 4 different 10X Chromium wells (see at the beginning of this section), we applied the aforementioned pipeline for each individual dataset. For running vireo, we looked for 4 and 5 different donors in the group1 and group2 replicates, respectively. Ultimately, we applied the function *vireoSNP.vcf.match_VCF_samples* from *vireo* package to match all the donors across the different replicates.

*Human mononuclear scRNA-seq analysis.* Viable (7AAD⁻) and metabolically active (calcein⁺) human muscle-derived mononuclear cells were FACS sorted and loaded in the 10X Chromium system. Libraries were generated following 10X Chromium Next GEM Single Cell 3' Reagent Kits v2 (CG00052 Rev F) and sequenced on the Illumina Hiseq2500 system. Sequencing reads were aligned using CellRanger 3.0.1 and the GRCh38.p10 annotation release 91. For QC, normalization, clustering and marker gene detection we followed the same workflow previously described in the snRNA-seq section. For the automatic mapping/annotation to Rubenstein et al. (GEO accession number: GSE130646)[24] and De Micheli et al. (GEO accession number: GSE143704)[23] datasets we used *SingleR* package (version 1.8.0)[33]. This method uses a Wilcoxon ranked sum test to compare the clusters of both datasets and then perform a Spearman rank correlation test to assign the labels of the reference dataset to our data. Then each cell label was projected into the original TSNE of our dataset.

For the integration analysis with the human adipose tissue dataset from Merrick et al. (GEO accession number: GSM3717979)[48], we first filtered out endothelial and smooth muscle cells from their dataset based on the markers provided by the authors (PECAM1 and ACTA2, respectively). We downsampled the adipose tissue dataset to an equal number of cells to avoid undesired results in the comparison due to sample size. Then, both adipose and muscle datasets were pooled and batch-effect corrected using *Harmony*[50] for downstream analysis. TSNE was calculated based on the dimensions generated by *Harmony*. Marker genes were calculated independently for each dataset and then intercrossed to generate the Euler diagrams.

For the integration analysis within *Tabula Sapiens*, we downloaded the TS_Fat and TS_Muscle datasets from figshare (https://doi.org/10.6084/m9.figshare.14267219.v4). For the fat dataset, we selected cells that met these parameters: donor:"TSP10", cell_ontology_class:"mesenchymal stem cell" or "fibroblast", method: "10X", compartment: "stromal", anatomical_information: "SCAT". For the muscle dataset, we applied these parameters: cell_ontology_class: "mesenchymal stem cell", method: "10X", anatomical_information: "Abdomen". Eventually, the final datasets were composed of 2393 and 3787 cells from fat and muscle tissues respectively. Subsequently, both datasets were integrated together following the same pipeline as before using *Harmony*.

*Mouse mononuclear scRNA-seq analysis.* Single cell suspensions were prepared as described below from mouse hindlimb muscle. Briefly hindlimb muscles were extracted, minced, and enzymatically digested. After erythrocyte lysis and filtering, debris was removed from the cell suspensions using a Debris Removal Solution (Miltenyi Biotec, 130-109-398) according to the manufacturer protocol. Following debris removal, dead cells were removed using a Dead Cell Removal Kit (Miltenyi Biotec, 130-090-101) according to the manufacturer protocol. This purified single-cell suspension was then loaded in the 10X Chromium system. Libraries were generated following 10X Chromium Next GEM Single Cell 3' Reagent Kits v2 (CG00052 Rev F) and sequenced on the Illumina Hiseq2500 system. Sequencing reads were aligned using CellRanger 3.0.1 and the GRCm38.p5 annotation release 91. For QC, normalization, clustering and marker gene detection we followed the same workflow as previously described in the snRNA-seq section. For downstream analysis, only FAPs were selected based on the differential expression of the canonical FAP markers *Pdgfra*, *Ly6a* (*Sca-1*), and *Cd34*. For the correlation heatmap, we selected differentially expressed genes from human FAPs clusters (calculated using *Seurat* package as described above) and performed a Pearson's correlation analysis in the mouse dataset to assess whether the FAPs markers correlate the same way as in the human sample.

For the integration analysis with the 10 weeks mouse adipose tissue dataset from Merrick et al. (GEO accession number: GSM3717978)[48], we first selected adipose progenitor cells based on the expression of the markers provided by the authors (*Ly6a*, *Cd34* and *Pdgfra*). Downsampling, pooling, batch-effect correction and marker gene detection (for Euler diagrams) were performed in the same way as in the human adipose-muscle integration (see above).

For the integration analysis with the 5 dpi glycerol dataset[53] (directly requested to the authors), we first processed the dataset following the method described in the original manuscript. Then, we selected FAPs based on the expression of the canonical markers *Dcn*, *Pdgfra* and *Cd34*. After selecting FAPs, we integrated and batch-corrected the data with our mouse muscle FAP dataset using *Harmony* as previously described. After integration we used, *TSCAN* package (version 1.34.0) for pseudotime analysis, selecting *Cd55*⁺ FAPs as the cluster of origin.

*Mouse bulk RNA-seq analysis.* PDGFRa⁺(eGFP⁺)MME⁺ and PDGFRa⁺(eGFP⁺) MME⁻ FAPs were freshly isolated and directly FACS sorted from mouse uninjured muscle into 700 μl RNA lysis buffer (Buffer RLT plus from RNeasy Plus Micro Kit (74034 QIAGEN). RNA extraction was performed using RNeasy Plus Micro Kit according to the manufacturer's protocol adjusting for the higher initial volume of RNA lysis buffer used. RNA quality was assessed by Agilent High Sensitivity RNA ScreenTape System (G2964AA). Only samples with RNA Integrity Number (RIN) ≥ 8.0 were further processed. Libraries were generated following the Smartseq II recommended protocol and were sequenced on the Illumina Novaseq 6000 instrument. Sequencing reads were pseudoaligned using Kallisto[74] and the GRCm38.p6 annotation release M23 to generate a count file matrix for each individual sample. Samples were pooled together (5 MME⁺ vs 5 MME⁻) on a single

dataset and genes with one count or less were filtered out. Following the DESeq2 analysis pipeline[75], variance stabilizing transformation (VST) for negative binomial data with a dispersion-mean trend was used for downstream analysis such as PCA and identification of highly variable genes. Differential expression analysis was performed on the raw counts after estimation of size factors (controlling for differences in the sequencing depth of the samples), the estimation of dispersion values for each gene and fitting a generalized linear model. Differentially expressed genes from DESeq2 pipeline were used for GSEA (fgsea package, see above) as well as the Hallmark and C2 curated gene sets (REACTOME subset) from MSigDB[30]. We performed GO enrichment analysis in the *molecular function* subontology, using the *enrichGO* function from *clusterProfiler* package (version 4.2.0)[76].

**Mouse FAP isolation.** For cell culture, flow cytometry, or bulk RNA sequencing, PDGFRa-eGFP mice were euthanized and all muscles from both hindlimbs were immediately dissected and placed in a Petri dish on ice. Muscles were minced with scissors until a homogeneous paste-like mash was formed. Thereafter, the mashed muscle was enzymatically digested in digestion buffer containing 0.2% Collagenase II (17101015, ThermoFisher Scientific), and 1.5% BSA (9048-46-8, PanReac AppliChem) in HBSS at 37 °C for 30 min with constant shaking. The reaction was stopped by adding an equal volume of TC media containing low glucose (1 g/L) DMEM supplemented with 10% FBS and the suspension was passed through a 100-μm cell strainer (#431751, Corning, New York, USA). Subsequently the suspension was centrifuged at 350 g and the supernatant discarded. The pellet was then suspended in 2 ml of erythrocyte lysis buffer (154 mM NH$_4$CL, 10 mM KHCO$_3$, 0.1 mM EDTA, pH = 7.35) and incubated on ice for approximately 1 min (longer for samples with more blood). 20 mL of cold PBS was then added to the suspension which was then immediately filtered through a 40-μm cell strainer (#431750, Corning, New York, USA) to remove tissue debris. Cell suspension was centrifuged at 550 g for 5 min at 4 °C. The pellet was then suspended in PBS/BSA (0.5% BSA) and incubated with anti-mouse MME antibody (Mouse Neprilysin/ CD10 Antibody, R&D Systems AF1126; 1:200), conjugated to the APC fluorophore using the Lightning Link APC Antibody Labeling Kit (705-0030, Novus Biologicals), for 1 h at 4 °C with shaking. Cell suspensions were then washed twice with PBS/BSA. For cell culture and bulk RNA sequencing, cells were brought immediately for FACS. For cell culture, eGFP$^+$ MME$^{-/+}$ cells were sorted on an SH800S sorter (Sony Biotechnology) using a 100 μm sorting chip into growth medium (DMEM low glucose, 20% FBS, 1% Penicillin-Streptomycin, and 5 ng/mL bFGF (PHG0266, ThermoFisher Scientific) and plated at a confluency no lower than 10,000 cells/cm$^2$. For bulk RNA sequencing, 15,000 eGFP$^+$ MME$^{-/+}$ cells were sorted by a FACS Aria III sorter (BD Bioscience) directly into Buffer RLT plus with β-mercaptoethanol from RNeasy Plus Micro Kit (74034 QIAGEN). For flow cytometry analysis, cells were then fixed in 2% PFA for 7 min on ice. After fixation the suspension was centrifuged at 550 g and the PFA removed. The cell pellet was then washed twice with PBS/BSA and resuspended for further detection of internal epitopes.

**Mouse cell culture.** Primary isolated FAPs were routinely cultured in FAP growth media containing DMEM Low glucose (22320022, ThermoFisher Scientific) supplemented with 20% fetal bovine serum (FBS) (10270-106, ThermoFisher Scientific), 1% Penicillin-Streptomycin (10,000 U/ml) (15140122, Thermo Fisher Scientific), and 5 ng/mL basic-FGF (PHG0266, ThermoFisher Scientific). Cells were expanded once before seeding for experiment unless otherwise noted. Full medium adipocyte differentiation was performed using white adipogenic induction medium containing DMEM high glucose (41966052, ThermoFisher Scientific) supplemented with 10% FBS, 1% Penicillin-Streptomycin, 0.5 mM IBMX (I5879, Sigma-Aldrich), 5 μM Troglitazone (T2573, Sigma-Aldrich), 0.25 μM dexamethasone (D2915, Sigma-Aldrich), and 5 ug/mL insulin (Sigma-Aldrich, I9278) during days 0–2. From days 2–7 cells were maintained in white adipogenic maintenance medium containing high glucose DMEM supplemented with 10% FBS, 1% Penicillin-Streptomycin, and 5 μg/mL insulin. Medium was changed every 48 h and collected on day 7 for imaging or RNA extraction. For low insulin adipogenic differentiation, cells were seeded at a density of 30,000 cells/well in a 96-well plate in FAP growth medium and allowed to attach overnight. Insulin was then added to the wells to a final concentration of 156 ng/mL. Medium was changed every 3 days before harvesting on day 5 for imaging or RNA extraction. For brown adipogenic differentiation, cells were induced from days 0–2 using brown adipogenic medium containing DMEM high glucose supplemented with 10% FBS, 1% Penicillin-Streptomycin, 5 ug/mL insulin, 1 nM T3 (T6397, Sigma-Aldrich), 1 μM rosiglitazone (AG-CR1-3570, Adipogen), 125 μM indomethacin (I8280, Sigma-Aldrich), 5 μM dexamethasone, and 0.5 mM IBMX. From days 2–10 cells were maintained in brown adipogenic maintenance medium containing high glucose DMEM supplemented with 10% FBS, 1% Penicillin-Streptomycin, and 5 μg/mL insulin, 1 nM T3, and 1 μM rosiglitazone. Medium was changed every 48 h and collected on day 10 for RNA extraction. Chondrogenic differentiation was performed using Mesenchymal Stem Cell Chondrogenic Differentiation Medium (with Inducers) (PromoCell Inc., c-28012) according to the manufacturer's protocol and harvested on day 21 for RNA extraction. Osteogenic differentiation was performed using Mesenchymal Stem Cell Osteogenic Differentiation Medium (PromoCell Inc., c-28013) according to the manufacturer's protocol and

harvested on day 14 for RNA extraction. Fibrogenic differentiation was performed using fibrogenic induction medium consisting of low glucose DMEM supplemented with 5% horse serum (HS, 16050-122, ThermoFisher Scientific), 1% Penicillin-Streptomycin, 5 ng/mL TGFβ (eBioscience, 14-8342-82) for 3 days when cells were harvested for RNA extraction.

**Mouse adipogenesis assays.** Ex vivo - Cells were seeded at a density of 30,000 c/ well in a 96 well plate in growth medium and allowed to attach overnight. Cells were then induced to differentiate as described above in either complete white adipogenic media or low insulin induction media. For GSK-3 inhibition, cells were induced to differentiate using low insulin medium supplemented with 1 μM CHIR99021 (4423, R&D Systems) or vehicle control (DMSO) throughout the entire differentiation protocol. After differentiation, cells were fixed with 4% PFA at 4 °C for 15 min. After washing twice with PBS, cells were stained with Hoechst (1:5000) and LD540 (100 ng/mL) in PBS for 30 min at room temperature protected from light. The staining solution was then removed and cells were washed 2 times with PBS. Cells were imaged using AxioObserver.Z1 fluorescence microscope (Carl Zeiss, Oberkochen, Germany).

In vivo – To determine adipogenic potential in vivo, isolated FAPs were injected into glycerol injected muscle. Contribution to fatty infiltration was determined 14 days after injection. PDGFRa-eGFP x Rosa26$^{mTmG}$ mice were used as donors and C57BL/6 J mice were used as recipient mice. Donors and recipients were injured using glycerol injection 3 days prior to isolation and injection. eGFP$^+$mT$^+$MME$^+$ and eGFP$^+$mT$^+$MME$^-$ cells were isolated as described above and sorted into FAP growth medium. Cells were washed twice in PBS to remove the growth medium and resuspended in PBS at a concentration of 1000 cells/μL. 20 μl of cell suspension was immediately injected into the recipient muscles. eGFP$^+$mT$^+$MME$^+$ and eGFP$^+$mT$^+$MME$^-$ cells were injected into separate legs of a single donor mouse. 14 days post injection, the muscles were harvested for immunofluorescence evaluation.

**Mouse FAP proliferation assays.** Ex vivo - To assess entry into the cell cycle, incorporation of 5-ethynyl-2'-deoxyuridine (EdU)(E10187, ThermoFisher Scientific) during the first 48 h in culture was assessed using the Click-iT Cell Reaction Buffer Kit (C10269, ThermoFisher Scientific) according to the manufacturer's instructions. Briefly, freshly sorted FAPs were seeded at 10,000 c/cm$^2$ in a 96- well plate in FAP growth media and allowed to attach for 12 h. Then, the media was changed to standard growth media containing 2.5 ug/mL EdU and refreshed once more 24 h later. Thereafter, the cells were washed once with PBS, fixed with 4% PFA for 15 min at 4 °C, and finally washed twice with PBS. To assess proliferation under standard culture conditions, cells were seeded at 5000 cells/well in a 96 well plate left to attach overnight. Media was then changed to FAP growth media containing 2.5 ug/ mL EdU and incubated for 2.5 h under standard growth conditions. Thereafter, the cells were washed once with PBS, fixed with 4% PFA for 15 min at 4 °C, and finally washed twice with PBS. After fixation, cells were permeabilized for 20 min at room temperature in 0.5% Triton X-100 with 3% BSA in PBS, then washed twice with 3% BSA in PBS and incubated with the Click-iT reaction cocktail for 45 min in the dark at room temperature. Thereafter, cells were briefly washed and counterstained with Hoechst (#62249, ThermoFisher Scientific). FAPs were imaged using an AxioObserver.Z1 fluorescence microscope (Carl Zeiss, Oberkochen, Germany).

In vivo – To assess proliferation in FAPs in vivo, mice were intraperitoneally injected with 1.25 mg EdU 6 h before tissue collection. After isolation and fixation as described above, cells were permeabilized in using 3% bovine serum albumin (BSA) 0.5% Saponin in PBS for 20 min at room temperature. After washing twice with PBS supplemented with 0.5% BSA (PBS/BSA), EdU was detected using the Click-iT™ Plus Picolyl Azide Toolkit (C10642, ThermoFisher Scientific, Massachusetts, USA) according to the manufacturer's instructions. To ensure eGFP fluorescence after incubation with the EdU reaction cocktail, 5 μl 100 mM CuSO$_4$ and 5 μl copper protectant were used in 500 μL reaction cocktail. After incubation with reaction cocktail, cells were stained with Hoechst (1:5000 in PBS), washed twice, and resuspended in PBS/BSA. Cells were brought for analysis on an LSRFortessa (BD Bioscience) cytometer.

**Detection of intracellular WNT signaling.** To detect intracellular WNT signaling via β-catenin, MME$^{-/+}$ were isolated as described above. Cells were seeded at 10,000 cells/cm$^2$ in FAP growth media in black bottom 96-well plates. Cells were allowed to attach for 48 h before fixation with 2% PFA at room temperature for 15 min. Cells were permeabilized using 0.3% Triton X-100 in PBS for 15 min. Blocking was then performed in blocking buffer (PBS + 3% BSA) for 30 min. Anti- non-phosphorylated (non-P) β-catenin primary antibody diluted 1:50 in blocking buffer was added and incubated for 2 h. Secondary antibody was added in blocking buffer and incubated for 1 h. Three five-min washes with PBS were performed between blocking, primary antibody incubation, and secondary incubation. Finally, cells were counterstained with Hoechst diluted 1:5000 in PBS, washed twice in PBS, and stored in PBS for imaging. Imaging was performed on an Olympus confocal microscope (FV1200). Nuclear localized non-P β-catenin was quantified as follows: The nuclear area was

determined on the image of the Hoechst channel and copied to the image of the non-P β-catenin channel. Area integrated density was then determined for that region of interest. Three areas of the background were then selected and the background fluorescence determined as the mean fluorescence of those three regions. Corrected total fluorescence = Integrated density – (Area of selected cell x mean fluorescence of background).

**In vivo apoptosis assay**. Mouse FAPs were isolated as described above. Cells were then permeabilized for 5 min at room temperature in permeabilization buffer (0.5% Saponin, 2% heat inactivated FBS (hi-FBS) (10500064, ThermoFisher Scientific) in DPBS (14190250, ThermoFisher Scientific)). Suspensions were then centrifuged at 550 g and the supernatant removed. Pellets were resuspended in biotinylated anti-cleaved caspase 3 (9654 S, Cell Signaling Technology) primary antibody diluted 1:500 in permeabilization buffer. Cells were washed twice with 2% hi-FBS in DPBS and resuspended in permeabilization buffer with PE-conjugated streptavidin and incubated for 45 min on ice. Cells were washed twice with 2% hi-FBS in PBS, resuspended in 0.5% BSA in PBS, and brought for analysis on an LSRFortessa (BD Bioscience) cytometer.

**Tissue immunofluorescence and histology**. Mouse tibialis anterior muscle samples and human muscle samples were harvested and embedded in OCT embedding matrix (6478.1, Carl Roth) and frozen in liquid N$_2$-cooled iso-pentane. 10 µm thick cryosections were then prepared at −26 °C to maintain adipocyte morphology. For evaluation of mTomato expression in intramuscular adipocytes, skeletal muscle cryosections were first taken without treatment for imaging of mTomato. After this, sections were fixed with 4% PFA for 10 min at RT and subsequently permeabilized using 0.3% Triton x-100 in PBS (PBST3). Then sections were blocked for 2 h at room temperature in PBS containing 25% normal goat serum (16-210-064, ThermoFisher Scientific) 0.3%Triton X-100 and 3% BSA. After blocking, sections were stained overnight at 4 °C using anti-Perilipin A primary antibody (P1998, Sigma-Aldrich) diluted 1:200 in perilipin staining buffer (PBST3, 10% normal goat serum, 3% BSA). Slides were subsequently washed with 0.03% Triton X-100 in PBS and then incubated with secondary antibody diluted 1:250 in perilipin staining buffer. For MME staining, sections were fixed in 4% PFA for 10 min at room temperature. After washing, slides were blocked in MME blocking buffer (10% Donkey Serum, 017-000-121, Jackson Immunoresearch) in PBS) for 1 h at room temperature. Anti-human MME antibody (AF1182, R&D Systems) was diluted 1:450 in MME blocking buffer and slides were incubated at 4 °C overnight. After washing, a secondary antibody was added at a concentration of 1:250 in MME blocking buffer and slides were incubated for 1 h at room temperature. For both protocols, the extracellular matrix was stained using wheat germ agglutinin (WGA) diluted 1:50 during secondary antibody incubation. Nuclei were detected using Hoechst diluted 1:5000 in PBS and for 5 min at room temperature. Slides were then washed 2 times with PBS and mounted using Immu-Mount (9990412, Thermo Fisher Scientific). Slides were imaged on an AxioObserver.Z1 fluorescence microscope (Carl Zeiss, Oberkochen, Germany). For detection of mTomato$^+$ Perilipin$^+$ adipocytes, images including Perilipin, WGA, and Hoechst staining were merged with mTomato containing images in Zen software using the "Add Channels" function.

H&E staining was performed as follows. Cryosections were submerged in Haematoxylin for 1 min followed by a wash in tap water. Subsequently slides were incubated in 70% ethanol for 3 min followed by 1 min incubation in Eosin. After, slides were incubated 3 times for 3 min in 100% ethanol. Finally, slides were incubated 2 times for 3 min in HistoClear (HS-200, National Diagnostics) and then mounted using Entellan (107960, Sigma-Aldrich). H&E images were acquired with an Eclipse Ti2 inverted microscope (Nikon) using a 10x objective.

**RNA extraction and quantitative RT-PCR**. RNA of FAPs and FAP derived adipocytes, chondrocytes, osteocytes, and fibroblasts was extracted using a RNeasy Plus Micro Kit according to the manufacturer's instructions (QIAGEN, 74034). RNA purity and concentration were assessed via a spectrophotometer (Tecan, Spark). RNA was reverse-transcribed to cDNA by High Capacity cDNA Reverse Transcription Kit (Thermo Fisher, 43-688-13). A SYBR Green-based master mix (ThermoFisher Scientific, A25778) was used for real-time qPCR analysis with primers listed in Supplementary Table 1. To compensate for variations in RNA input and efficiency of reverse-transcription, Rna18s5 was used as a housekeeping gene. The delta-delta CT method was used to normalize the data.

**Statistics and reproducibility**. All images in the figures are representative of the data and the staining quality. All bar graphs represent mean ± SEM unless otherwise specified. GraphPad Prism software (version 9.2) was used for statistical analysis. When comparing two independent group means, a Student's t test (two tailed, unpaired) was used. When comparing group means of paired measurements (e.g., MME- vs MME+ cells from the same mouse) a Student's t test (two tailed, paired) was used. For experiments evaluating more than one variable, a two-way ANOVA with Tukey's multiple comparison was used. For experiments evaluating more than one variable with a repeated measures design, a two-way ANOVA with Šidák's multiple comparison test was used. Each figure legend indicates the statistical approach for each experiment displayed in the figure. Asterisks in figures denote statistical significance. No experiment-wide multiple test correction was applied. R (version 4.1.2) and related packages (see Transcriptomic analysis section) were used for RNA-seq statistical analysis. Unless otherwise indicated, the default statistical test (e.g., Wilcoxon Rank Sum test for Marker gene detection) from each package was used (specified in *Transcriptomic analysis* section).

**Reporting summary**. Further information on research design is available in the Nature Portfolio Reporting Summary linked to this article.

## Data availability

All sequencing data (single-nuclei, single-cell, and bulk RNA-seq) reported in this study are available at the Gene Expression Omnibus (GEO) repository under the accession number GSE200487. Please check https://shiny.debocklab.hest.ethz.ch/Fitzgerald-et-al/ for data visualization. All other data are available from the corresponding author on request.

Source data for all the figures are provided in a single excel file as Supplementary Data 1.

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

## Acknowledgements

We thank Veronique Juvin from SciArtWork (https://sciartwork.com/) for help with Fig. 4J, and the Functional Genomics Center Zürich (FGCZ) as well as the ETH Flow Cytometry Core Facility (E-FCCF) for excellent technical support. We thank the Swiss Center for Musculoskeletal Biobanking for support during human sample collection and processing. This project was funded by ETH Zurich Research Grant ETH-16 17-1. KDB is endowed by the Schulthess Foundation and supported by a European Research Council (ERC) Starting Grant (716140). Part of this work was funded through the Swiss National Science Foundation (310030_176056).

## Author contributions

G.F. conceptualized the study, designed and performed the experiments, and wrote the paper. G.T. performed and analyzed all sn-, sc- and bulk RNA-seq experiments and wrote the paper. T.G., I.S.A., J.Z., E.M. assisted in experiments and data analysis. N.C.C. and N.A.M. supervised the human study. M.L. acquired human samples and supervised the study. R.S. performed and analyzed MRI based quantification of human fat infiltration. J.F. performed ex vivo human FAP experiments. K.D.B. conceptualized the study, acquired funding and wrote the paper. All authors approved the final version of the manuscript.

## Competing interests

The authors declare no competing interests.
