## [Peer Review File · Communications Biology]

This manuscript has been previously reviewed at another Nature Portfolio journal. This document only contains reviewer comments and rebuttal letters for versions considered at Communications Biology.

Reviewers' comments:

Reviewer #1 (Remarks to the Author):

The authors have adequately addressed my concerns raised about the original manuscript and provided additional data that substantiates their observation of distinct FAP populations with an MME+ population showing increased adipogenic potential. Of particular note, the study nicely incorporates comparative analyses of these populations in murine and human samples which will be of value to FAP/fibroblast community.

Minor comment – gene names on lines 303 and 304 should be italicized.

Reviewer #2 (Remarks to the Author):

Overall, the authors have responded to all comments except for the following comment:

- MME+ FAPs increase both in CTX and Gly injuries. When do they start to be adipogenic? It looks like MME marks a progenitor population that also gives rise to fibrotic FAPs (author's response: As shown in Figure S6 in the revised manuscript, in basal conditions MME+ FAPs show upregulation of *Cebpa*, which is an early pro-adipogenic regulator. This is consistent with our bulk RNAseq experiment where we saw upregulation of the HALLMARK_ADIPOGENESIS pathway in MME+ FAPs. With this in mind, it seems that MME+ FAPs are more primed for adipogenesis than their MME- counterparts in basal conditions. Nonetheless, when analyzing single cell data from glycerol injected muscle at 5dpi, we observed that some 5 dpi FAPs already begin to express mature adipogenic markers such as *Pparg*, *Plin1*, and *Adipoq*. This we did not observe in the MME+ FAPs. In fact, 5dpi (glycerol injury) FAPs reduce MME expression before switching on adipocyte markers. Additionally, we also detected reduced MME expression in adipocytes in the human single nuclei dataset (Figure 1C). Thus, we hypothesize that MME+ FAPs might be more primed for adipogenesis than their MME- counterparts, while maintaining the capacity to differentiate into other lineages (Figures S5I-L).

As the reviewer noted, we did see pathways related to the regulation of extracellular matrix upregulated in the MME+ FAPs in our bulk RNAseq experiment (Figure 5C, left panel). Although this was the case, we did not see increased propensity to differentiate into fibroblasts as determined by expression of *Acta2* after fibrogenic differentiation (Figure S5I). @CommunicationsBiology: we have included the pseudotime analysis in the revised manuscript.)

The additional experiments reinforce the fact that MME+ FAPs have an adipogenic potential, higher than the MME- FAPs. However, it is still not clear if the MME+ population have the double fibro-adipogenic potential. RTqPCR using sorted MME+ cells after glycerol and cardiotoxin injury on fibrogenic genes such as *Col3a1*, *Col1a1*, *Postn*... should be provided. This should be addressed in the results and discussion accordingly.

Additional comment:

Response to Reviewer #1, comment 3:

" Although MME+ FAPs are more highly adipogenic under minimal adipogenic induction, when isolated, brought in culture, and induced to differentiate into other mesenchymal lineages, they still possess differentiation capacity in the fibrogenic (*Acta2*) (Figure S5I), osteogenic (*Bglap2*, *Sox9*, *Comp*) (Figure S5J), and chondrogenic (*Sox9*, *Comp*) (Figure S5K) lineages."

Sox9 and *Comp* are not osteogenic markers

RESPONSE LETTER

Reviewer #1 (Remarks to the Author):

The authors have adequately addressed my concerns raised about the original manuscript and provided additional data that substantiates their observation of distinct FAP populations with an MME+ population showing increased adipogenic potential. Of particular note, the study nicely incorporates comparative analyses of these populations in murine and human samples which will be of value to FAP/fibroblast community.

We thank the reviewer for their positive evaluation of our work.

Minor comment – gene names on lines 303 and 304 should be italicized.

We have adapted this in the manuscript.

Reviewer #2 (Remarks to the Author):

Overall, the authors have responded to all comments except for the following comment:

- MME+ FAPs increase both in CTX and Gly injuries. When do they start to be adipogenic? It looks like MME marks a progenitor population that also gives rise to fibrotic FAPs (author's response: As shown in Figure S6 in the revised manuscript, in basal conditions MME+ FAPs show upregulation of *Cebpa*, which is an early pro-adipogenic regulator. This is consistent with our bulk RNAseq experiment where we saw upregulation of the HALLMARK_ADIPOGENESIS pathway in MME+ FAPs. With this in mind, it seems that MME+ FAPs are more primed for adipogenesis than their MME- counterparts in basal conditions. Nonetheless, when analyzing single cell data from glycerol injected muscle at 5dpi, we observed that some 5 dpi FAPs already begin to express mature adipogenic markers such as *Pparg*, *Plin1*, and *Adipoq*. This we did not observe in the MME+ FAPs. In fact, 5dpi (glycerol injury) FAPs reduce MME expression before switching on adipocyte markers. Additionally, we also detected reduced MME expression in adipocytes in the human single nuclei dataset (Figure 1C). Thus, we hypothesize that MME+ FAPs might be more primed for adipogenesis than their MME- counterparts, while maintaining the capacity to differentiate into other lineages (Figures S5I-L). As the reviewer noted, we did see pathways related to the regulation of extracellular matrix upregulated in the MME+ FAPs in our bulk RNAseq experiment (Figure 5C, left panel). Although this was the case, we did not see increased propensity to differentiate into fibroblasts as determined by expression of *Acta2* after fibrogenic differentiation (Figure S5I). @CommunicationsBiology: we have included the pseudotime analysis in the revised manuscript.)

The additional experiments reinforce the fact that MME+ FAPs have an adipogenic potential, higher than the MME- FAPs. However, it is still not clear if the MME+ population have the double fibro-adipogenic potential. RTqPCR using sorted MME+ cells after glycerol and cardiotoxin injury on fibrogenic genes such as *Col3a1*, *Col1a1*, *Postn*... should be provided. This should be addressed in the results and discussion accordingly.

The proposed experiments for the fibrogenic genes are highly interesting and we thank the reviewer for this suggestion. However, they are difficult to conduct. As we wrote in the manuscript, MME+ cells lose MME before acquiring adipogenic potential. We confirmed this using the pseudotime analysis (also in the paper), so we cannot use published or own scRNAseq datasets to address this. The only possible way we could decently approach this question is to perform additional experiments where we transplant fluorescently labeled MME+ and MME- FAPs into glycerol as well as cardiotoxin injured mice and sort the cells after 5 days to investigate their gene activation pattern. Not only does such an

experiment require a lot of mice, but we are also currently unable to execute it since we do not have ethical permission to inject FAPs into cardiotoxin injured muscle. Acquiring this ethical permission unfortunately would take months in Switzerland. Consequently, we have included the lack of in vivo data on the fibrogenic potential as a limitation of the study in the corresponding section of the manuscript.

Additional comment:

Response to Reviewer #1, comment 3:

” Although MME+ FAPs are more highly adipogenic under minimal adipogenic induction, when isolated, brought in culture, and induced to differentiate into other mesenchymal lineages, they still possess differentiation capacity in the fibrogenic (*Acta2*) (Figure S5I), osteogenic (*Bglap2*, *Sox9*, *Comp*) (Figure S5J), and chondrogenic (*Sox9*, *Comp*) (Figure S5K) lineages.”

Sox9 and Comp are not osteogenic markers

We have removed *Sox9* and *Comp* from the osteogenic panel and we performed an additional qPCR to assess the expression of *Runx2* after osteogenic differentiation (revised Figure S5J). Both MME^{+/-} FAPs showed a slight increase in *Runx2* expression after 14 days of osteogenic stimulation in an similar extent to previous studies (<https://doi.org/10.1186/1471-2474-11-104>).

REVIEWERS' COMMENTS:

Reviewer #2 (Remarks to the Author):

Although the authors have addressed most comments, 2 corrections should still be made:

1- The authors should clearly state in the discussion that they cannot completely exclude the bi-potentiality (fibrotic and adipogenic) of the MME+ FAPs, and that further analyses would be required to address this bi-potentiality.

2-Line 304: Sox9 and Comp are not osteogenic markers and are indeed listed as chondrogenic markers. Please replaced with appropriate markers.